# Chirality in Action: Time-Aware Video Representation Learning by Latent Straightening

**Piyush Bagad**   **Andrew Zisserman**

VGG, Dept. of Engineering Science, University of Oxford

## Abstract

Our objective is to develop compact video representations that are sensitive to visual change over time. To measure such time-sensitivity, we introduce a new task: chiral action recognition, where one needs to distinguish between a pair of temporally opposite actions, such as "opening *vs*. closing a door", "approaching *vs*. moving away from something", "folding *vs*. unfolding paper", *etc*. Such actions (i) occur frequently in everyday life, (ii) require understanding of simple visual change over time (in object state, size, spatial position, count . . . ), and (iii) are known to be poorly represented by many video embeddings. Our goal is to build time aware video representations which offer linear separability between these chiral pairs. To that end, we propose a self-supervised adaptation recipe to inject time-sensitivity into a sequence of frozen image features. Our model is based on an auto-encoder with a latent space with inductive bias inspired by *perceptual straightening*. We show that this results in a compact but time-sensitive video representation for the proposed task across three datasets: Something-Something, EPIC-Kitchens, and Charade. Our method (i) outperforms much larger video models pre-trained on large-scale video datasets, and (ii) leads to an improvement in classification performance on standard benchmarks when combined with these existing models.

## 1  Introduction

The ever-increasing scale of video content on the Internet demands efficient and compact descriptors that can be readily used for classification, ranking and search. The goodness of video descriptors (or video representations) is largely measured in terms of action recognition on standard benchmarks such as Kinetics-400 [11], UCF101 [75] and Something-Something [28] to name a few. While action recognition performance provides a reliable single measure for the representation, more insight is obtained by establishing how well a given video descriptor encodes various aspects of the video such as objects, scene context, motion and temporal dynamics. We can coarsely categorize these aspects into: *static* properties (objects, scene, *etc*) and *dynamic* properties (motion, visual change, *etc*). It is well established that, apart from Something-Something, most contemporary video benchmarks tend to focus more on static properties [10, 35, 44, 49, 108]. While there has been an effort to shift the focus to evaluating dynamic [28, 49, 59, 73], properties of actions are still entangled with static understanding without a clear definition of dynamics. In this work, our objective is to study a specific time-sensitive property: understanding how well video descriptors encode *visual change* in a video.

What do we mean by "understanding" visual change? Consider an example action pair: "a person climbing up a ladder" vs. "a person climbing down a ladder". In this case, the vertical position of the person changes over time and it is temporally opposite in the two actions. An ideal video descriptor should encode this change and use it distinguish between such action pairs. We call such action pairs *'chiral'*, and the task of distinguishing between them *'chiral action recognition'*. Where can we find such actions? These are quite common in everyday life, and humans effortlessly recognize them.

39th Conference on Neural Information Processing Systems (NeurIPS 2025).

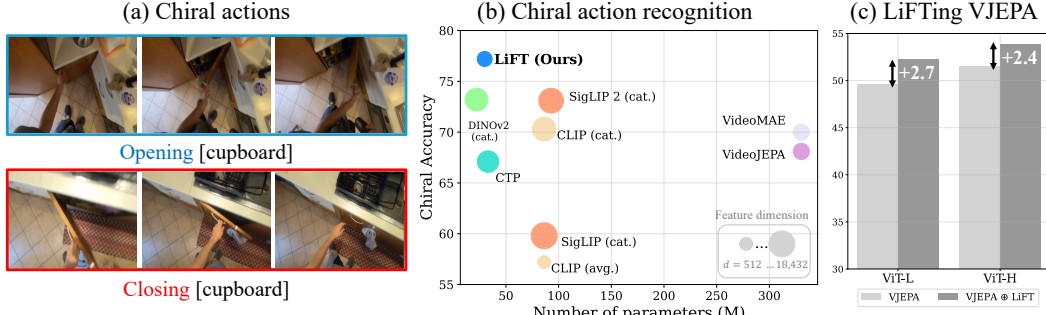

Figure 1: (a) We introduce *chiral actions*: temporally opposite action pairs to test time-awareness of video descriptors. We build a meta-dataset of chiral actions by mining SSv2, EPIC and Charades. To build time-aware descriptors, we propose LiFT (Linearized Feature Trajectories) that disentangles a sequence of DINOv2 features into a *static* and *dynamic* descriptor. (b) LiFT outperforms contemporary video and image models in recognizing chiral actions across the three datasets. Moreover, LiFT descriptors can be plugged into existing models (VideoJEPA or VideoMAE) to improve action recognition on standard benchmarks. (c) shows a linear probe of LiFT with VideoJEPA on SSv2.

In this work, we mine chiral pairs from three existing datasets (SSv2 [28], EPIC-Kitchens [17], and Charades [74]), to set up a chiral evaluation benchmark.

Turning to the descriptors, most existing video representations are obtained by two classes of methods, either (i) self-supervised video embeddings – natively multi-frame models, trained on millions of videos [5, 79, 89], or (ii) per-frame image models adapted for video data that are usually trained for specific datasets [57, 64]. We borrow from both lines of work and propose a self-supervised adaptation recipe that yields general video descriptors that outperform much larger models [5, 89] for chiral action recognition. Specifically, we hypothesize that a representation will be time sensitive if the per-frame features form a smooth trajectory in latent space. Inspired by Perceptual Straightening [32], we operationalize this by learning a model that maps per frame features from a strong image model to ordered points on lines in latent space. We show that the two vectors representing these high-dimensional lines yield time-aware video descriptors. We call our model *LiFT* for Linearized Feature Trajectories. Qualitatively, we show that LiFT learns compact video descriptors that encode a smooth, continuous approximations of the feature trajectories. Quantitatively, we show that LiFT descriptors are time-aware: they can distinguish between chiral action pairs across three datasets without specialized fine-tuning.

While LiFT descriptors achieve strong results on chiral action recognition, can they be more generally useful, say by combining with other video models? In this spirit, we evaluate linear and attentive probes with LiFT descriptors combined with video models such as VideoJEPA [5] on four standard action recognition datasets: Kinetics-400 [11], UCF-101 [75], HMDB-51 [42] and SSv2 [28]. We show that the combination of LiFT and a given video model always outperforms solely using the video model, across different video models, across all four benchmarks. This demonstrates that the time-sensitivity in LiFT preserves information that is complementary to standard video models, which in turn helps *lift* performance on action recognition benchmarks. In summary, our contributions are:

1. We propose a new task called chiral action recognition which requires discounting static context and accounting for the dynamic change in a video. We formulate a meta-dataset from three action recognition datasets to benchmark this task.

2. We propose LiFT: a self-supervised recipe to adapt DINOv2 features into a compact, time-sensitive, and general video descriptor. LiFT outperforms much larger video models (e.g., $10\times$ bigger and trained on over $6\times$ samples) by over 7% on the proposed chiral benchmark.

3. We demonstrate that LiFT encodes time-sensitive information that is complementary to contemporary video models. We show that combining LiFT descriptors with video models such as VideoJEPA [5], VideoMAE [79] and InternVideo2.5 [89] lifts performance with linear as well as attentive probes as compared to only using these models.

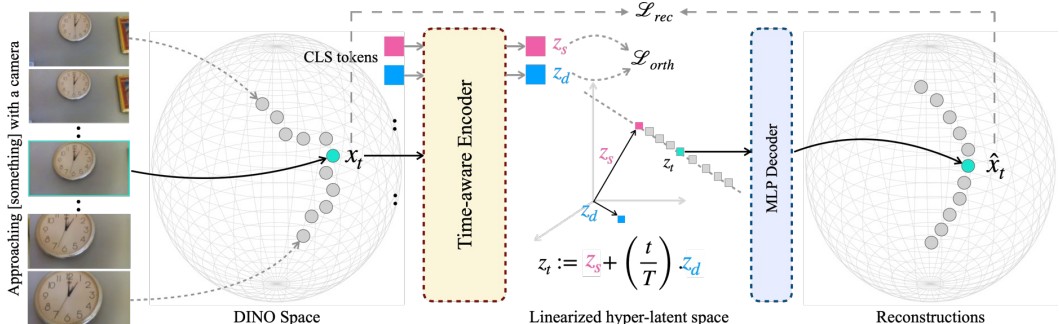

Figure 2: **Linearized Feature Trajectories (LiFT).** We propose LiFT as a simple adaptation of image features to obtain time-aware video descriptors. First, we encode each frame independently with a DINOv2 backbone. Then, we pass them through a Transformer encoder with two learnable tokens: $\mathbf{z}_s$ (static) and $\mathbf{z}_d$ (dynamic). Next, inspired by Perceptual Straightening [32], we enforce linearity in the latent space which enables reconstruction of the trajectory with only the two learnable tokens. We do not show position encodings and projection layers for brevity. The network is trained with the usual reconstruction loss and an orthogonality regularization between the static and dynamic tokens. Once the model is trained, an input video is represented by concatenation of $\mathbf{z}_s, \mathbf{z}_d$.

## 2   LiFT: Video Representation by Linearized Feature Trajectories

Our objective is to learn a single time-aware descriptor vector for a given video. We want to avoid training large video models [5, 79] from scratch given an academic compute budget. In such a scenario, usually Parameter-Efficient Fine-Tuning (PEFT) is employed to adapt image models for video data [57, 58]. However, PEFT is dataset-specific supervised tuning while our goal is to obtain a more general video descriptor while still adapting an image model without any label supervision.

Considering these desiderata, we propose a simple recipe based on adapting a strong image model with reconstruction in the latent space. Our central hypothesis is that for videos that depict a visual change, the per-frame features lie on smooth trajectories that encode such change over time. While these trajectories tend to be non-linear, we can map them to a latent space in which they are parametrized by a line. This is loosely inspired by the Perceptual Straightening Hypothesis [32]: humans perceive image sequences that are non-linear in pixel space as straight lines in the perceptual space. Thus, a video can be represented simply by the vectors that define this line in the latent space. Owing to this linear formulation, we call the model *LiFT*: Linearized Feature Trajectories.

The schematic diagram of LiFT is shown in Fig. 2. Formally, consider a video as a sequence of images $\{I_t\}_{t=1}^T, I_t \in \mathbb{R}^{C \times H \times W}, \forall t$. First, we encode each frame independently using an image model $\Phi$ . We only retain the global CLS token per frame.

$$\mathbf{x}_t = \Phi(I_t) \in \mathbb{R}^D, \forall t. \tag{1}$$

Then, we train an autoencoder network to reconstruct $\{\mathbf{x}_t\}_1^T$ while learning meaningful video descriptors in its latent space. Now, we describe the Encoder-Decoder network and how we train it.

**Encoder.**    The encoder takes in sequence of frame features and outputs a descriptor for the sequence. First, we project the feature sequence to a potentially lower-dimensional space.

$$\mathbf{e}_t = P_{\downarrow}(\mathbf{x}_t) \in \mathbb{R}^d, \forall t. \tag{2}$$

Sinusoidal position encoding is added to encode the frame index. Then, a Transformer Encoder takes in this sequence $\{\mathbf{e}_t\}_1^T$ along with two learnable CLS-like tokens: $\mathbf{e}_s$ and $\mathbf{e}_d$ that are used to encode *static* and *dynamic* information respectively in the feature sequence.

$$\mathbf{z}_s, \mathbf{z}_d = \text{TransformerEncoder}\left(\mathbf{e}_s, \mathbf{e}_d, \{\mathbf{e}_t\}\right) \tag{3}$$

We collect these tokens output by the Transformer, denoted by $\mathbf{z}_s$ and $\mathbf{z}_d \in \mathbb{R}^d$. The overall video descriptor is given by their concatenation $\mathbf{z} \in \mathbb{R}^{2d}$.

**Decoder.** The decoder takes in $\mathbf{z}_s, \mathbf{z}_d$ and time index $t$ and outputs feature vector as time $t$. In the latent space, we enforce a linearity constraint defined by $\mathbf{z}_s, \mathbf{z}_d$ as shown in Fig. 2.

$$\mathbf{z}_t := \mathbf{z}_s + \left(\frac{t}{T}\right).\mathbf{z}_d \in \mathbb{R}^d, \forall t \tag{4}$$

The decoder is a two-layer MLP that takes in $\mathbf{z}_t$ and outputs the reconstructed feature at time $t$.

$$\hat{\mathbf{x}}_t = \mathrm{MLPDecoder}(\mathbf{z}_t) \in \mathbb{R}^D, \forall t. \tag{5}$$

**Training objective.** We train the network with the usual reconstruction loss and a regularizer that encourages orthogonality between the static and dynamic latent vectors.

$$\mathcal{L} := \mathcal{L}_{\mathrm{rec}} + \lambda \mathcal{L}_{\mathrm{orth}} = \sum_{t=1}^{T} \|\mathbf{x}_t - \hat{\mathbf{x}}_t\|_2^2 + \lambda . \cos\text{-sim}\left(\frac{\mathbf{z}_s}{\|\mathbf{z}_s\|_2}, \frac{\mathbf{z}_d}{\|\mathbf{z}_d\|_2}\right) \tag{6}$$

Note that this is unsupervised. Once trained, we discard the Decoder and use the Encoder to get a LiFT video descriptor $(\mathbf{z}_s, \mathbf{z}_d)$.

**Time-awaremess of LiFT descriptors.** To gain an insight into what LiFT learns, first, we visualize the joint tSNE embeddings of the original and reconstructed trajectories on sample videos in Fig. 3(a). We also visualize the tSNE embeddings of an example action pair in Fig. 3(b). These illustrate that: (a) LiFT outputs a smooth, continuous approximation of the original trajectories evident from the tSNE plot. In a sense, LiFT captures the "*arc of change*" depicted in the video; and, (b) Thanks to the simple linearization, LiFT is compelled to learn compact descriptors that can distinguish between temporally opposite actions such as "opening vs closing a door". Note that the linearization is only applied in the latent space, thus, LiFT's reconstructed trajectories can be non-linear.

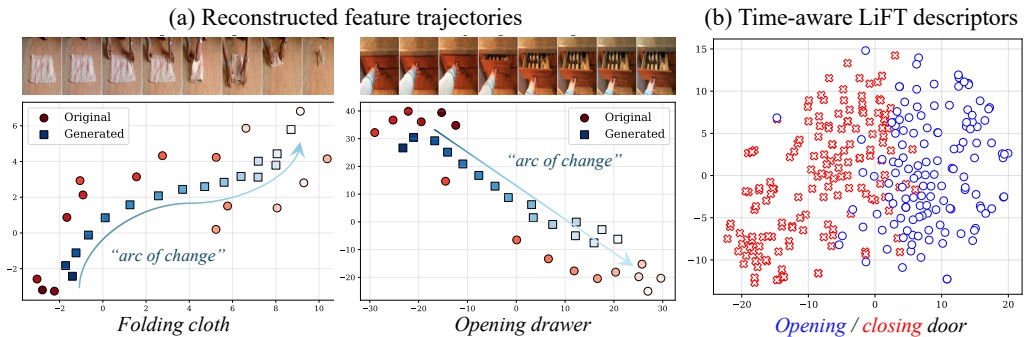

Figure 3: **Qualitative analysis.** (a) LiFT reconstructs a smooth, continuous approximation of the original feature trajectories roughly encoding the arc of visual change. (b) LiFT descriptors are time-aware: LiFT distinguishes between temporally opposite actions such as "opening/ closing door".

**Implementation details.** We use DINOv2 (ViT-S/14) [56] with registers [18] as the base image feature extractor with dimension $D=384$. We only use the CLS token output for each frame. We linearly sample $T=16$ frames from each video and compute the features ahead of training. The input feature sequence is first projected to the space of the Encoder $\mathbb{R}^D \to \mathbb{R}^d$; we choose $d=384$. The Encoder is a standard Transformer [14] with 4 layers and 8 attention heads each. The Encoder uses sinusoidal position encoding [14] to encode the frame index. The outputs from the two CLS tokens, $\mathbf{z}_s, \mathbf{z}_d \in \mathbb{R}^d$ are then projected to a space where we impose the linearity constraint: $\mathbb{R}^d \to \mathbb{R}^d$. The Decoder is an MLP with 2 hidden layers each followed by a GeLU activation [34] and LayerNorm [2]. Overall, the model has $8.7M$ trainable parameters beyond the $22M$ parameters of the frozen DINOv2 encoder. We provide more details on the architecture in the Supplemental. We also provide ablations varying $d$ and the amount of training data in the Supplemental.

**Training.** We train LiFT on Kinetics-400 [11] which has about 240K videos. Since the image encoder is frozen, we pre-compute features which makes the training very efficient. The model is trained for 500 epochs with a batch size of 128 with Adam optimizer [41] with a learning rate of 0.001 and a LRPlateau scheduler.

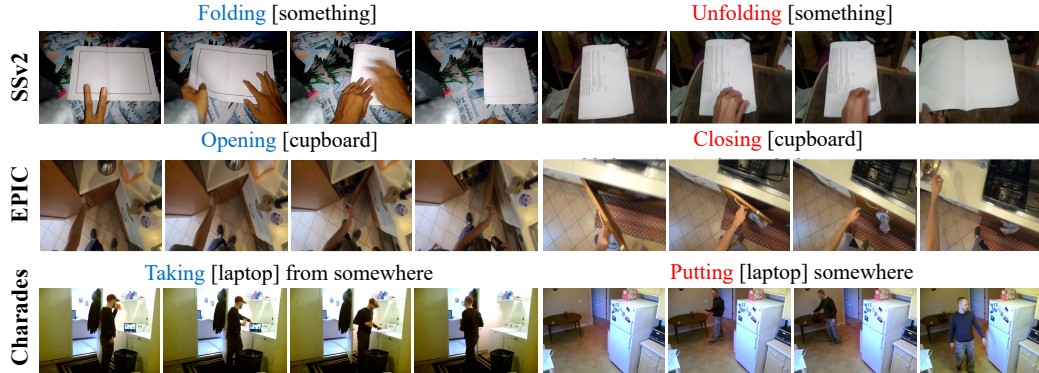

Figure 4: **Example chiral pairs** from each of the three datasets. To distinguish the actions in each pair, one needs to discount the spatial context and account for *what* is changing over time and *how*.

## 3 Chirality in Action (CiA) Dataset

To quantitatively measure time-awareness of LiFT (or other video descriptors), we propose a new task, namely, *chiral action recognition*. The study of time has covered aspects such as arrow of time [60, 91], order of frames [53, 102], recognizing temporally fine-grained actions [73, 104], or space-time tracking [96]. Prior work has developed specialized benchmarks and models for such tasks. In contrast, we want to measure time-awareness of a general video descriptor in recognizing simple everyday actions. It is well-established that existing action recognition benchmarks are biased to spatial understanding [35, 72]. Thus, we narrow down our focus on what we call *chiral actions*.

**Chiral actions.** In daily life, we often perform actions such as "closing/opening a door", "folding/unfolding a cloth" or "getting in/out of a car". Our proposal is that a good video model should distinguish between such temporally opposite actions. Loosely inspired by the notion of *visual chirality* [50], we call such action pairs as *chiral*, *i.e.*, pairs that are approximately mirror reflections along the time dimension. Consider the examples shown in Fig. 4. In distinguishing between these actions, one needs to discount the spatial context and account for the visual change over time.

Note that unlike in the study of arrow of time [91], we do not artificially reverse time-arrow but in a sense, our chiral actions have a naturally opposite arrow of time. The notion of chiral actions is also closely related to *reversible* actions studied in [62]. However, our work is complementary since we can use the methods in [62] to identify chiral actions and then evaluate video models on them. Finally, chiral actions, as we define them, are similar to *nearly symmetric actions* introduced in concurrent work by Ponbagavathi and Roitberg [61]. However, we build a meta-dataset of a more general mix of datasets that includes a richer set of actions, has both exo- and ego-centric videos and is larger in size.

**Constructing CiA dataset.** We build a meta-dataset out of chiral subsets of popular action recognition datasets. We identify three datasets to build a benchmark for chiral action recognition: Something-Something (SSv2) [28], EPIC-Kitchens (EPIC) [17], and Charades [74]. These datasets come with action recognition labels with separate verb and noun annotations. For each dataset, we build chiral pairs from the provided labels as follows.

1. We pass the list of action verbs to ChatGPT and ask it to find antonym pairs. We manually verify the output to remove pairs that have hallucinated verbs or those that are not visually antonymous.
2. For each verb pair, we group similar nouns together. For example, for verb pair "opening" vs "closing", nouns such as "door/cupboard/drawer" that represent visually similar actions are grouped. This group is represented by the triplet ("opening", "closing", "[door]"). Likewise, ("opening", "closing", "[box]") represents a separate chiral group where objects such as "tiffin/box/parcel", *etc* are grouped together. Thus, each chiral group is a triplet consisting of a pair of opposite verbs and the associated noun.
3. For each chiral group, we split the videos into train and test sets following the split defined in the original dataset.

Some basic numbers for each dataset are provided in Table 1 and visual examples are shown in Fig. 4.

| Base dataset | Chiral groups | Avg videos/group | Example chiral group |
|---|---|---|---|
| Something-Something (SSv2) [28] | 16 | 852.8 | Folding / Unfolding [something] |
| EPIC-Kitchens (EPIC) [17] | 66 | 412.2 | Opening / Closing [door] |
| Charades [74] | 28 | 768.4 | Taking / Putting a [laptop] |

Table 1: **Numbers for CiA meta-dataset.** We mine chiral action pairs in three existing action recognition datasets to build our benchmark. Visual examples are shown in Fig. 4.

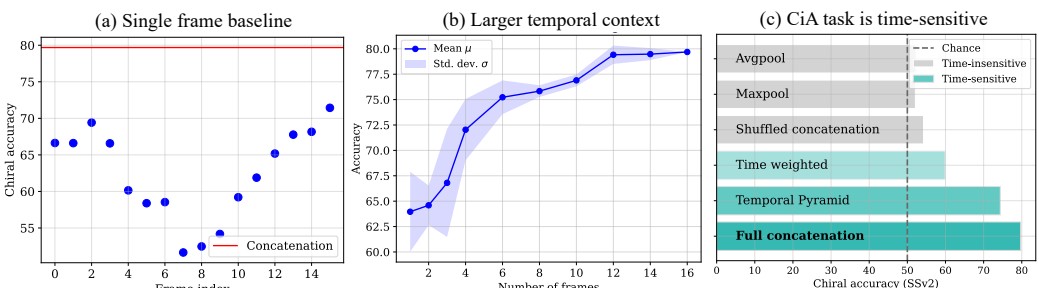

Figure 5: **Time-sensitivity of chiral action recognition.** On SSv2 (Chiral), we run simple baselines with DINOv2 features to test how time-sensitive the task is. In (a), (b), the video descriptor is obtained by concatenating per frame features. (a) Single frames chosen at the ends tend to do well but lag far behind using all frames. (b) Increasing number of frames provides consistent benefit while saturating around 16 frames. (c) Time insensitive pooling (*e.g.*, mean) of features is noticeably worse than time-sensitive pooling (*e.g.*, full concatenation or time-weighting).

**Evaluation protocol.** We measure time-sensitivity as a test of linear separability of chiral actions. For a given video model and a chiral group, we extract video representations for each video from the two antonym classes. We train a linear classifier in the feature space on the train set. We repeat this for each chiral group and report the average accuracy on test set across all groups. There are several reasons to choose this evaluation protocol: (i) We want the evaluation method to be as simple as possible so that the true strength of the representation is measured without confounding with the strength of the evaluation model (*i.e.*, in this case, a linear probe); (ii) Since there can be several chiral groups (e.g., $K{=}16$ in SSv2), it is computationally convenient to train simple linear probes rather than a more complicated and compute-heavy model for each group; (iii) Evaluating frozen features is much more common and practical [5, 12] as models get larger and evaluations need to be faster.

**Properties of chiral action recognition.** To analyze the time-sensitivity of chiral action recognition, we consider simple pooling baselines for DINOv2 features. The simplicity of the model enables us to study the task while discounting model strength. We show the results on SSv2 in Fig. 5. This shows that (a) A single frame chosen at either ends of the video yields a decent baseline as the end frames depict either the start or the end state of an action which is sometimes informative. However, the performance of such single frame baseline significantly lags using more frames as shown in (b). Increasing the temporal context with more frames consistently improves performance. (c) Finally, we compare time-insensitive pooling methods (*e.g.*, average) with time-sensitive ones (*e.g.*, full concatenation). We find that time-sensitive pooling does significantly better. Overall, this analysis highlights that the task at hand (chiral action recognition) benefits from time-aware ordering of many frames which establishes that it does not suffer as much from a single- or static-frame bias [35, 72].

## 4 Experiments

In this section, we first present results on our proposed chiral action recognition task. Then, we explore more general action recognition tasks where our descriptor is useful. In particular, in Section 4.1, we show that LiFT outperforms much larger video models in distinguishing between temporally opposite actions without specialized fine-tuning; and in Section 4.2, we show that plugging the LiFT descriptor with other general video models lifts their performance on standard action recognition benchmarks.

### 4.1 Chiral action recognition

**Experimental details.** We follow the evaluation protocol detailed in Section 3. For each chiral group, we compute video descriptors as described in Section 2 and train a linear classifier. We compare

against two sets of baselines. (i) image models such as the image encoder in CLIP. (ii) Video models trained with self- or language supervision on large-scale video datasets. Typically, we sample $T{=}16$ frames linearly from the video and compute a single descriptor. For image models, we concatenate the per-frame features to obtain the video descriptor. For video models based on R(2+1)D architecture (e.g., TCLR [19]), we sample $T' = 8$ clips over the span of the video and concatenate clip features to obtain a single descriptor. For Transformer-based video models, if the model uses a CLS token, we treat that as descriptor or average pool all the output tokens unless stated otherwise.

**Main result.** From the results shown in Table 2, we observe that the proposed LiFT features achieve the best performance on SSv2, EPIC and Charade, while being compact ($d = 768$). Notably, LiFT beats much heavier video models such as VideoMAE, VideoJEPA and InternVideo2.5. Interestingly, naively concatenating image features with a strong model (DINOv2, SigLIP2) generally performs better or at par with much heavier video models (e.g., VideoMAE) reinforcing that the complete sequence of image features does retain rich information for chiral action recognition. However, naturally, naively concatenating yields very bulky descriptors which may be impractical, say, in indexing a database with millions of videos. Less surprisingly, Transformer-based video models outperform ResNet (R(2+1)D) based models.

Finally, we find that careful feature pooling can make a notable difference. For example, Intern-Video2.5/VideoMAE/VideoJEPA output a sequence of tokens per frame. Average pooling over space and concatenating over time does much better than average pooling over space and time. Nevertheless, average pooling output tokens (e.g., from VideoMAE) is still reasonable and time sensitive because (i) it does not have an explicit CLS token as a single video embedding, (ii) it comprises of 3D space-time tokens enhanced by temporal position encoding; different frames interact with each other in every layer of the transformer.

| Model | Architecture | Pooling | D ↓ | Chiral Accuracy ↑ | | |
|---|---|---|---|---|---|---|
| | | | | SSv2 | EPIC | Charades |
| Chance | - | - | - | 50.0 | 50.0 | 50.0 |
| *Image models with naive concatenation* | | | | | | |
| CLIP [65] | ViT-B/16 | Average | 512 | 53.5 | 63.4 | 54.7 |
| CLIP [65] | ViT-B/16 | Concat. | 8192 | 71.6 | 71.6 | 67.7 |
| BLIP2 [45] | ViT-B/16 + Q-Former | Concat. | 12288 | 73.2 | 70.3 | 67.3 |
| SigLIP [103] | ViT-B/16 | Concat. | 18432 | 57.9 | 66.2 | 55.2 |
| SigLIP 2 [80] | ViT-B/16 | Concat. | 12288 | 76.8 | 74.7 | 67.8 |
| DINOv2 [56] | ViT-S/14 | Concat. | 6144 | 79.7 | 74.1 | 65.8 |
| *Image models adapted for video (trained on Kinetics-400)* | | | | | | |
| ST-Adapter [57] | ViT-B/16 | Learned | 768 | 50.5 | 63.7 | 54.4 |
| DiST [64] | ViT-B/16 | Learned | 512 | 52.1 | 59.9 | 55.9 |
| *Video models* | | | | | | |
| Tubelet Contrast [77] | R(2+1)D | Concat. | 4096 | 64.6 | 62.8 | 58.9 |
| TCLR [19] | R(2+1)D | Concat. | 4096 | 67.9 | 62.5 | 58.8 |
| CTP [86] | R(2+1)D | Concat. | 4096 | 78.8 | 64.4 | 58.0 |
| VideoMAE [79] | ViT-L/16x16x2 | Average | 1024 | 80.3 | 70.5 | 59.1 |
| VideoMAEv2 [87] | ViT-B/16x16x2 | Average | 768 | 65.3 | 67.5 | 55.5 |
| SIGMA [69] | ViT-B/16x16x2 | Average | 768 | 66.5 | 69.1 | 56.1 |
| MME [76] | ViT-B/16x16x2 | Average | 768 | 78.4 | 70.8 | 57.5 |
| VideoJEPA [5] | ViT-L/16x16x2 | Average | 1024 | 80.4 | 67.4 | 56.4 |
| InternVideo 2.5 [89] | InternViT-6B | Average | 4096 | 55.8 | 66.1 | 55.4 |
| VideoMAE [79] | ViT-L/16x16x2 | Time concat. | 8192 | 85.7 | 75.0 | 66.1 |
| VideoJEPA [5] | ViT-L/16x16x2 | Time concat. | 8192 | 85.4 | 70.8 | 57.1 |
| InternVideo 2.5 [89] | InternViT-6B | Time concat. | 32768 | 80.0 | 70.9 | 62.8 |
| LiFT (Ours) | ViT-S/14 | Learned | 768 | **86.6** | **75.5** | **69.5** |

Table 2: **Results on chiral action recognition.** (1) Our method (LiFT) of efficiently adapting sequential information in DINOv2 features has the best performance on the chiral splits of all three datasets. (2) On average, sequence of image features contain stronger discriminative information for chiral actions in comparison to native video models.

**Ablation study.** We run ablation over key design choices such as the image encoder in LiFT. Results of ablation study on SSv2 are shown in Table 3. We note that self-supervised image features such as iBOT [107] and DINOv2 [56] outperform language-supervised models such as CLIP [65] or SigLIP2 [80]. We hypothesize that since DINOv2/iBOT are better at capturing spatial details, their feature trajectories of a video tend to better capture smooth visual change compared to language-supervised models. From the last two rows, we also establish that using the orthogonal loss does provide a small benefit in chiral action recognition.

**What kinds of change are easier to understand?** We categorize the visual change involved in each chiral action pair. For example, in "moving towards/away from the camera", the object size or depth changes or in "taking/putting one of many objects on table", the object count changes. We average the performance across all chiral pairs that depict a given kind of visual change across all three datasets and report in Table 4. We find that LiFT features shine in distinguishing chiral actions that involve change in object state or count but struggle in those with change in position along $x$-axis.

| Image encoder | Architecture | SSv2 (Chiral) |
|---|---|---|
| CLIP | ViT-B/16 | 75.9 |
| SigLIP2 | ViT-B/16 | 77.8 |
| BLIP2 | ViT-B/16 + QF | 75.7 |
| iBOT | ViT-B/16 | 80.3 |
| DINOv2 | ViT-B/14 | 85.4 |
| DINOv2 | ViT-L/14 | 85.9 |
| LiFT w/o $\mathcal{L}_{\mathrm{orth}}$ | ViT-S/14 | 85.9 |
| LiFT | ViT-S/14 | **86.6** |

Table 3: **Ablation on image encoders.** (i) self-supervised image features (*e.g.*, iBOT/DINOv2) outperform language-supervised features (*e.g.*, CLIP), (ii) with DINOv2, features out of larger models do not necessarily show improvement, (iii) using orthogonality loss helps.

| Change type | VMAE | VJEPA | DINO | LiFT |
|---|---|---|---|---|
| Dist. bet. objects | 70.8 | 87.5 | 83.3 | 87.5 |
| Object count | 64.2 | 62.4 | 69.5 | 72.4 |
| Object size/depth | 96.8 | 96.8 | 92.2 | 96.8 |
| Object state | 72.9 | 66.3 | 75.9 | 80.7 |
| Spatial position $\leftrightarrow$ | 96.3 | 96.1 | 75.7 | 75.2 |
| Spatial position $\updownarrow$ | 91.5 | 89.7 | 79.7 | 93.6 |

Table 4: **Performance across kinds of change.** Color green denotes performance at par or better than competing models and red denotes worse. LiFT features shine in distinguishing chiral actions that involve change in object state or count but struggle in those with change in position along $x$-axis. DINO here denotes naive concatenation of DINO features.

## 4.2 LiFTing video models on standard benchmarks

While LiFT outperforms much heavier video models in recognizing chiral actions, can it help improve performance on standard action recognition? We conduct an extensive linear probe evaluation across four standard datasets: Kinetics-400 (K400) [85], UCF-101 [75], HMDB-51 [42] and SSv2 [28]. The experimental details are provided in the Supplemental. As shown in Table 5a, while LiFT by itself does not beat top video models, concatenating LiFT with such video models consistently lifts their performance. This indicates that LiFT descriptors have complementary information. Furthermore, in Table 5b, we ablate over kinds of probes and model sizes used. We consistently observe a benefit with LiFT. Interestingly, with VideoJEPA as well as VideoMAE, ViT-L combined with LiFT even outperforms a scaled up ViT-H. Thus, overall, an adapter such as LiFT when combined with standard video models can provide strong video representations useful for classification, retrieval and search.

## 4.3 Investigating performance difference between vertical vs. horizontal motion

As noted in Table 4, there is a substantial difference between the ability of LiFT's to distinguish vertical translation motion ($\uparrow$ vs. $\downarrow$) and horizontal motion ($\rightarrow$ vs. $\leftarrow$). We uncover two reasons for this. First, the base model is better at encoding changes along vertical axis compared to horizontal axis. This is shown in Table 4 by comparing LiFT with a baseline with concatenated DINOv2 features over time. The latter is better at encoding $\updownarrow$ changes compared to $\leftrightarrow$ changes. Second, and more substantially, this difference is caused by the anisotropic sensitivity of DINOv2 feature sequence, i.e., DINOv2 features vary less with horizontal movement vs vertical movement. To quantify this, we generate synthetic sequences with a checkerboard background and a colored disc that moves either horizontally or vertically at a constant rate. As the measure, we compute the variance over time and then average it across the feature dimensions. This time variance in vertical shift sequences is about 25% higher than that in horizontal shift sequences. Finally, we demonstrate three tricks to reduce this gap and improve overall performance: (i) scaling up image resolution, (ii) using a better base

| Model | K400 | UCF | HMDB | SSv2 |
|---|---|---|---|---|
| Chance | 0.25 | 0.99 | 1.96 | 0.58 |
| LiFT | 55.4 | 86.6 | 65.2 | 30.8 |
| VJEPA | 59.8$^\dagger$ | 91.3 | 76.1 | 49.6$^\dagger$ |
| VJEPA ⊕ LiFT | 63.7 | 92.6 | 78.0 | 52.3 |
| Δ | +3.9 | +1.3 | +1.9 | +2.7 |
| VideoMAE | 55.0 | 83.6 | 66.5 | 38.3 |
| VideoMAE ⊕ LiFT | 63.6 | 88.8 | 72.6 | 46.3 |
| Δ | +8.6 | +5.2 | +6.1 | +6.0 |
| InternVid2.5 | 62.8 | 88.2 | 71.9 | 23.4 |
| InternVid2.5 ⊕ LiFT | 65.9 | 90.3 | 75.3 | 35.9 |
| Δ | +3.1 | +2.1 | +3.4 | +11.5 |

(a) LiFT combined with video models lifts their performance results across four action recognition benchmarks.

| Arch. | Probe | Base | Base⊕LiFT | Δ |
|---|---|---|---|---|
| *VideoJEPA* | | | | |
| ViT-L | Non-lin. | 51.7 | 54.2 | +2.5 |
| ViT-L | Attentive | 65.9$^\dagger$ | 66.9 | +1.0 |
| ViT-L | Linear | 49.6$^\dagger$ | 52.3 | +2.7 |
| ViT-H | Linear | 51.5 | 53.9 | +2.4 |
| *VideoMAE* | | | | |
| ViT-L | Non-lin. | 43.9 | 50.1 | +6.2 |
| ViT-L | Attentive | 61.5 | 63.7 | +2.2 |
| ViT-S | Linear | 19.4 | 37.3 | +17 |
| ViT-B | Linear | 25.6 | 41.1 | +16 |
| ViT-L | Linear | 38.3 | 46.3 | +6.0 |
| ViT-H | Linear | 40.0 | 46.9 | +6.9 |

(b) LiFT consistently improves performance of video models across model scale and probes.

Table 5: **Results on standard action recognition datasets.** LiFT improves probing accuracies with standard video models across datasets and model sizes. $^\dagger$Note: The numbers for VideoJEPA are obtained with our experimental setup. We could not precisely reproduce the numbers reported in the paper [5] even using their codebase. We have reached out to the authors for clarification.

model (WebSSL [22]), and (iii) using test-time rotation augmentation. More details are provided in the Appendix C.4.

## 5   Related Work

**Human perception of videos and time.** Psychologists have tried to understand how humans perceive visual change in videos (*e.g.*, motion) for a long time [83, 90]. More recently, Hénaff et al. [32] present a remarkable finding: visual system in humans and macaques transforms complex pixel dynamics in videos into straighter temporal trajectories [32, 33]. Straighter trajectories make predictions easier and predictions are a fundamental part of human perception. Inspired by this insight, we learn an auto-encoder on image feature trajectories with linearity baked in the latent space. Although there is some prior work inspired by Perceptual Straightening [27, 31, 55], we apply it to a sequence of image features and show that it leads to more general time-aware representations.

**Time-aware video representations.** Temporally pooling image sequences has been a classical way of representing videos. Carefully crafted pooling in pixel space [8], in motion/flow space [9] and in embedding spaces [23] have been devised. Since the prominence of deep learning on videos, time has been creatively used as a source of self-supervision: space-time jigsaw [40], time arrow [62, 91], time order [26, 94, 97], speed [6], tracking [36, 85],contrasting temporal views [19, 63, 68], cycle consistency in time [21] or explicitly modeling temporal dynamics [15, 37, 105]. Modern video encoders are based on Transformers [1, 5, 7, 51, 79, 87]. Data-efficiency [77, 79] and time-sensitivity [20, 69, 78, 98] of video models continue to be active areas of research [71]. In this work, we investigate time-sensitivity of existing models through chiral action recognition and propose a simple recipe to embed videos based on summarizing trajectories of image features. Concurrent to our work, Xue et al. [95] propose a reinforcement learning-based training strategy to instill arrow of time awareness in video LLMs. Our chiral actions are related to the arrow of time, but we do not artificially reverse the arrow of time in a video; instead we aim to distinguish actions that are naturally opposite along the arrow of time.

**Action recognition benchmarks.** Early datasets for action recognition in videos include UCF-101 [75], HMDB-51 [42], Sports-1M [38] and Kinetics [11]. Transformers [84] prompted the rise of multimodal video datasets with text [4, 52], audio [13, 25] and 3D [29, 81]. LLMs led to the rise of instruction-tuning datasets [46, 106] and benchmarks [24, 47, 59] for videos. However, the community has repeatedly discovered that a majority of these do not actually test for time; a single frame or an unordered set of frames would suffice to recognize the action in the video [10, 16, 35, 44, 49, 108]. SSv2 [28], Diving-48 [49] introduced temporally sensitive actions while other datasets evaluate specific aspects: causal/counterfactual reasoning [59, 93, 99], compositionality [30, 101], concept-binding [39, 70], temporal prepositions [3] and verbs [54, 72]. In this work, we propose chiral action recognition that evaluates video features in discriminating temporally opposite actions. Our definition

of chirality is related to that of equivariant actions in Price and Damen [62] but their aim was more to discover actions invariant/equivariant to time flipping. It is also related to the subset of action antonyms in Cores et al. [16]. Chirality is also related to *nearly symmetric actions* in concurrent work by Ponbagavathi and Roitberg [61]. However, unlike [61], we propose a more general video embedding model trained in an unsupervised manner.

**Efficient adaptation of image models to videos.** Given the computational cost of training video models from scratch, Parameter Efficient Fine-Tuning (PEFT) methods to adapt image models for videos have emerged [48, 57, 64, 67, 82]. Since we use frozen DINOv2 features, our work also adapts image model for video recognition. However, PEFT methods are usually trained separately for each downstream dataset and generally used in a supervised learning setup. Our method is more generally applicable. It is trained in an unsupervised manner on Kinetics-400 and the resulting video embeddings are shown to be applicable for chiral action recognition across three datasets.

## 6 Discussion and Conclusion

In an effort to develop time-sensitive video descriptors, we proposed Linearized Feature Trajectories (LiFT): a simple recipe to adapt DINO per-frame features with an auto-encoder with an inductive bias inspired by Perceptual Straightening [32]. As a measure of time-sensitivity, we introduce chiral action recognition to distinguish between temporally opposite actions such as "opening vs. closing a door". We created the CiA meta-dataset with chiral pairs mined from three public datasets: SSv2 [28], EPIC [17], and Charades [74]. On CiA, we show that LiFT outperforms much heavier video models including VideoJEPA [5] and VideoMAE [79] while being compact. Furthermore, we show that the time-sensitive LiFT descriptors contain information that is complementary to standard video models. For example, LiFT when combined with VideoJEPA lifts performance across four action recognition benchmarks: Kinetics [11], UCF [75], HMDB [42] and SSv2 [28].

**Future work.** Since we only use per frame CLS tokens, LiFT likely misses out on some spatial details, especially horizontal translation as shown in Table 4. Investigating ways of mitigating this, *e.g.*, using a sequence of dense feature maps, is an open avenue for future research. Furthermore, since our recipe is self-supervised, combining it with other compute-heavy self-supervised pre-training paradigms such as Masked Modeling [5, 79] or Autoregression [66, 92] should be interesting avenues to imbue more time-sensitivity into these representations.

**Societal impact.** Our work improves video representations for temporal actions thus impacting video applications, *e.g.*, indexing a massive video database or text to video retrieval. A less direct impact can be on improving domestic robots that frequently need to perform temporal actions like opening/closing a door. Our work also has a positive impact in terms of energy efficiency: it is much more efficient in training and inference compared to larger video models. Finally, our work inherits the risks of any video models in being applied to NSFW video content.

**Acknowledgments.** This research is funded by the EPSRC Programme Grant VisualAI EP/T028572/1, and a Royal Society Research Professorship RSRP\R\241003. We thank Ashish Thandavan for support with infrastructure, and Sindhu Hegde and Makarand Tapaswi for useful discussions. We also thank the anonymous reviewers that helped improve this work.

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

# A Dataset: Chirality in Action (CiA)

**Metadata and examples.** In Table 6, we show the chiral groups constructed in SSv2. Similarly, we construct 66 chiral groups in EPIC and 28 groups in Charades. In Table 7, we show the number of videos in the chiral splits of each of the three datasets. We attach the chiral splits for all three datasets as part of the Supplemental. We also provide a single CSV file that includes the chiral groups for all three combined data sets. We also show more examples of chiral pairs from each of the three datasets in Fig. 6, Fig. 7 and Fig. 8. In general, since SSv2 has single canonical actions, it is a cleaner test bed for chiral action recognition. EPIC and Charades usually have a more cluttered visual context where cues for chiral recognition are more subtle.

| Verb → | Verb ← | Noun (object) |
|---|---|---|
| Pulling [something] from left to right | Pulling [something] from right to left | ['something'] |
| Pushing [something] from left to right | Pushing [something] from right to left | ['something'] |
| Turning the camera left while filming [...] | Turning the camera right while filming [...] | ['something'] |
| Approaching [something] with your camera | Moving away from [something] with your camera | ['something'] |
| Closing [something] | Opening [something] | ['object'] |
| Closing [something] | Opening [something] | ['door'] |
| Closing [something] | Opening [something] | ['bottle'] |
| Closing [something] | Opening [something] | ['book'] |
| Closing [something] | Opening [something] | ['purse'] |
| Closing [something] | Opening [something] | ['drawer'] |
| Moving [...] and [...] away from each other | Moving [...] and [...] closer to each other | ['something'] |
| Moving [something] away from the camera | Moving [something] towards the camera | ['something'] |
| Moving [something] down | Moving [something] up | ['something'] |
| Putting [something similar to other things ...] | Taking [one of many similar things on the table] | ['something'] |
| Turning the camera downwards while filming [...] | Turning the camera upwards while filming [...] | ['something'] |
| Folding [something] | Unfolding [something] | ['something'] |

Table 6: **Chiral groups in SSv2.** We construct 16 chiral groups in SSv2 by identifying temporally opposite verbs. Note that "opening vs. closing" is split across different objects representing entirely different actions. Noun "['something']" denotes a placeholder which can include any appropriate object that fits with the action verb.

| Dataset | Chiral groups | Total videos | | Avg. videos per chiral group | | Avg. duration (s) |
|---|---|---|---|---|---|---|
| | | Train | Validation | Train | Validation | |
| SSv2 | 16 | 12216 | 1430 | 763.5 | 89.4 | 3.6 |
| EPIC | 66 | 24101 | 3108 | 365.1 | 47.1 | 1.6 |
| Charades | 28 | 16018 | 5498 | 572.1 | 196.4 | 8.6 |

Table 7: **CiA dataset size.** For each of the constituent datasets, we show the total number of videos in the proposed chiral split and also the average number of videos per chiral group. Note that we train one linear probe for each chiral group.

**Time-sensitivity of CiA.** In Fig. 9 and Fig. 10, we repeat the experiments to check time-sensitivity (Fig 5 in the main paper) of the CiA benchmark on all three datasets. Our inferences about time-sensitivity hold for all three datasets.

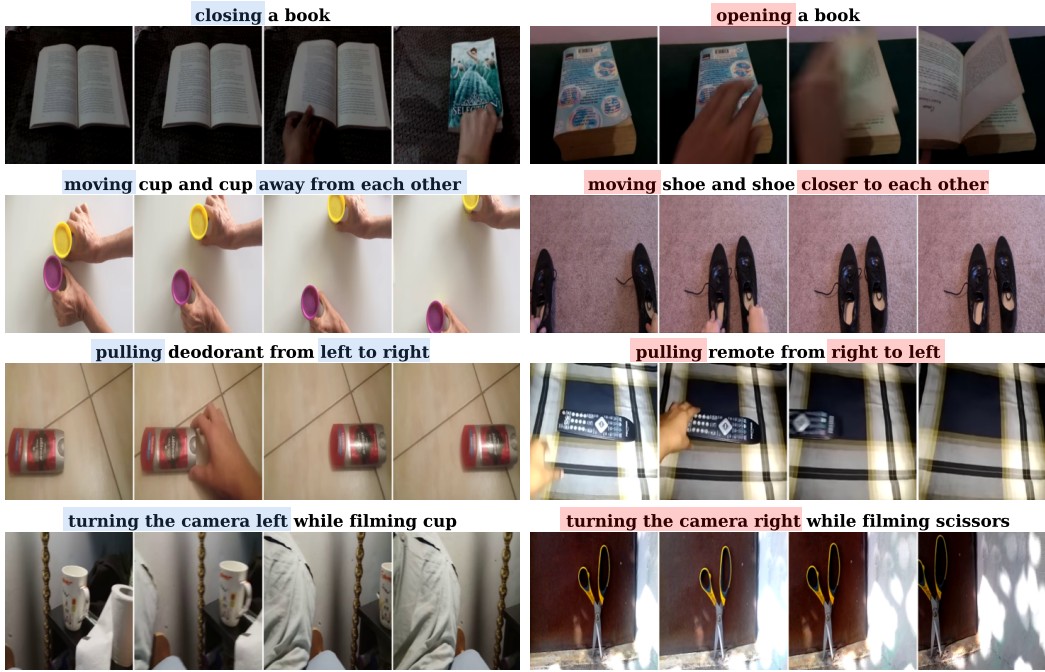

Figure 6: **CiA samples from SSv2.** More examples of chiral pairs from the train set of SSv2. The positive direction actions are marked in blue while the negative direction ones are marker in red.

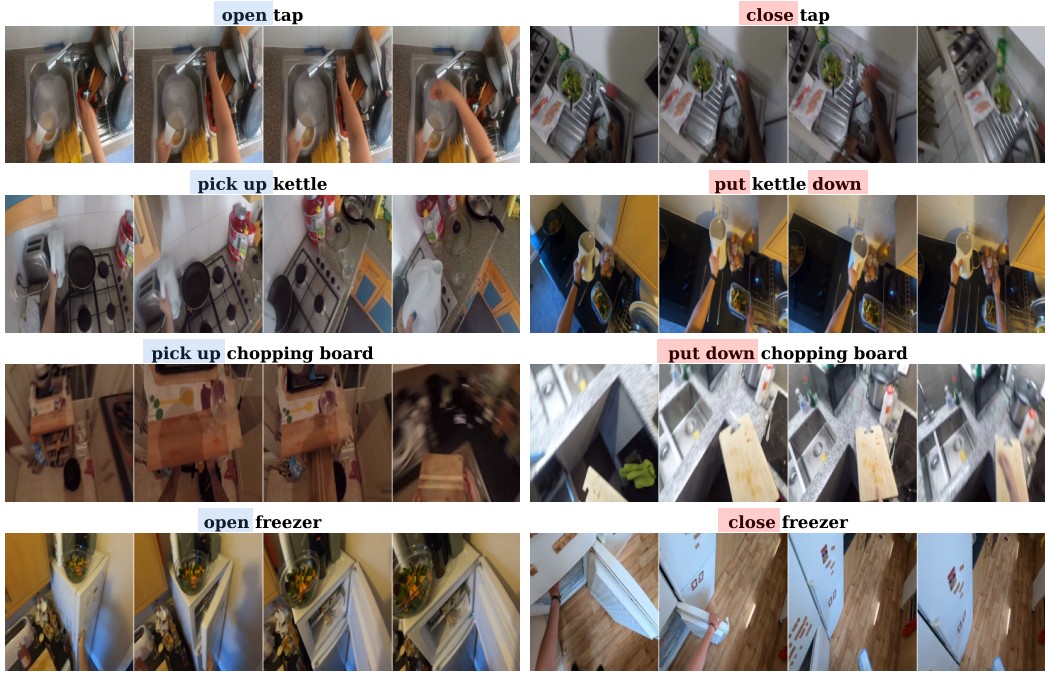

Figure 7: **CiA samples from EPIC.** More examples of chiral pairs from the train set of EPIC. The positive direction actions are marked in blue while the negative direction ones are marker in red.

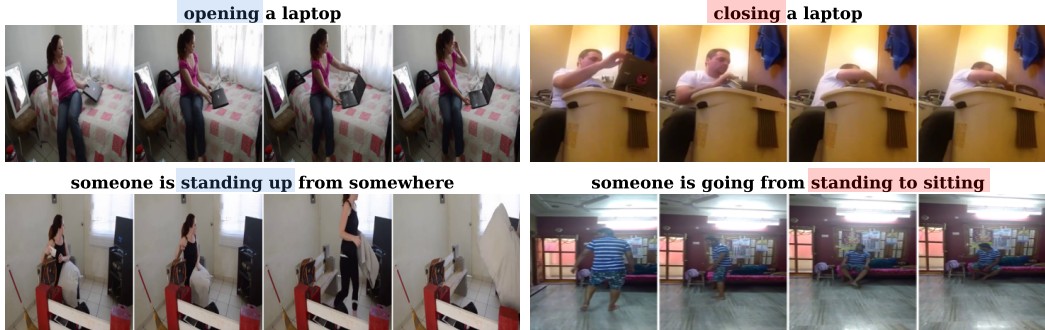

Figure 8: **CiA samples from Charades.** More examples of chiral pairs from the train set of Charades. The positive direction actions are marked in blue while the negative direction ones are marker in red.

## B    Model: Linearized Feature Trajectories (LiFT)

**Architecture.**    A detailed sketch of the architecture is provided in Fig. 11. In LiFT, the Encoder takes in a sequence of features $\{\mathbf{x}_t\}_t$ and outputs two descriptor tokens $\mathbf{z}_s, \mathbf{z}_d$. First, a linear layer projection is applied $\mathbb{R}^D \to \mathbb{R}^d$. Then, Sinusoidal position encoding is added representing frame index $t$. The two CLS tokens $\mathbf{e}_s, \mathbf{e}_d$ are initialized randomly. Then, the CLS tokens along with the sequence tokens are passed through a Transformer with $L{=}4$ blocks and $H{=}8$ heads each. Each block has a multi-head self-attention (MHSA) layer followed by a FFN layer. Both the layers are preceded by LayerNorm layers. Then, the outputs for the two CLS tokens are projected with a linear layer ($\mathbb{R}^d \to \mathbb{R}^d$) followed by LayerNorm. This gives the latent descriptors $\mathbf{z}_s$ and $\mathbf{z}_d$.

The decoder takes in $\mathbf{z}_s, \mathbf{z}_d, t$ and outputs $\hat{\mathbf{x}}_t$. First, we construct an intermediate representation for the frame at index $t$ using our linearity constraint in the latent space.

$$\mathbf{z}_t = \mathbf{z}_s + (t/T).\mathbf{z}_d \tag{7}$$

Then, this is passed to an MLP network with two hidden layers each followed by GeLU activation and LayerNorm. The first hidden layer maps $\mathbb{R}^d \to \mathbb{R}^{2d}$ and the second layer maps $\mathbb{R}^{2d} \to \mathbb{R}^{2d}$. This is followed by a linear projection ($\mathbb{R}^{2d} \to \mathbb{R}^D$) back to the DINOv2 space.

**Compute resources.**    In order to train LiFT, we first compute and store feature vectors for DINOv2 ViT-S/14. This feature computation is run on 4 NVIDIA RTX A4000 16GB GPUs in parallel. It takes about 12 GPU hours to compute features for 250K videos in Kinetics-400. Once features are computed, LiFT is trained on a single consumer-grade GPU (*e.g.*, NVIDIA RTX A4000, Tesla P40, Quadro RTX 8000, NVIDIA RTX A6000). A single training run takes about 15 GPU hours.

**Computational cost comparison to video models.**    LiFT has a total of 30M parameters (22M frozen DINOv2 parameters and 8M trainable parameters) trained on 240K videos. In contrast, native video models, e.g., Video JEPA (smallest variant is ViT-L) has about $10\times$ the parameters (305M) and training samples (2M). Furthermore, during inference, for a clip with $T{=}16$ frames with resolution $224 \times 224$ on the same hardware, LiFT takes 2.3 ms/clip while VJEPA takes 8.5 ms/clip. Thus, overall, LiFT provides a compact, fast, and complementary video embedding when compared to native video models.

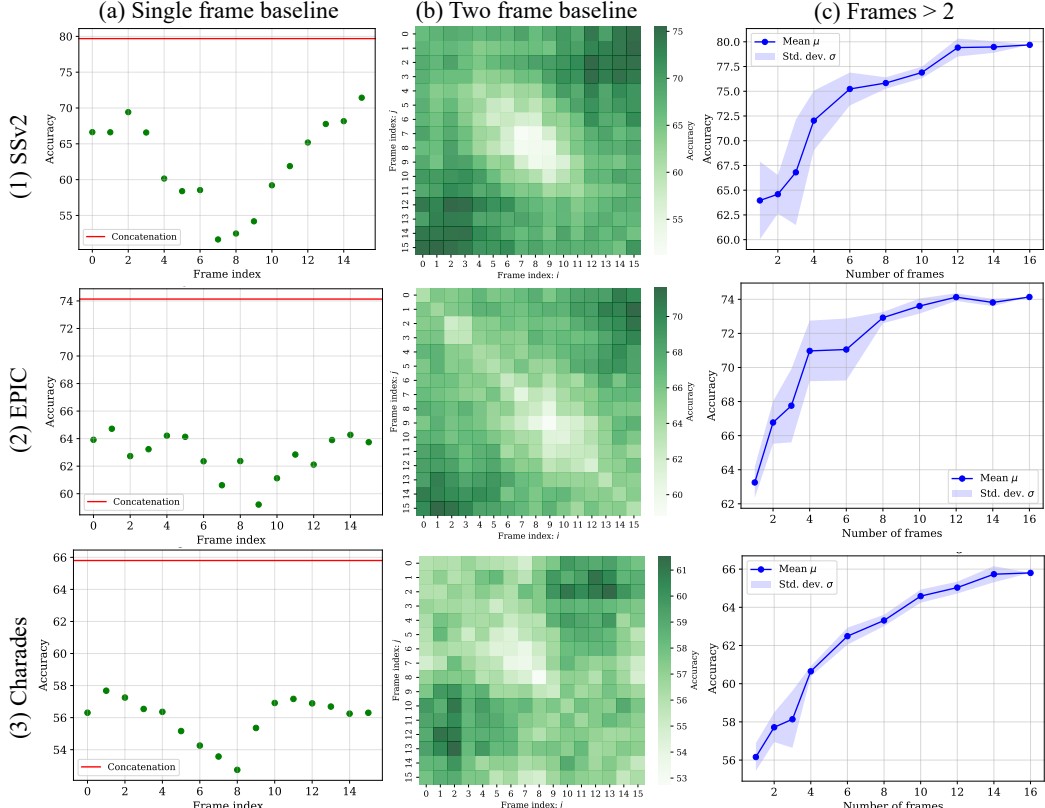

Figure 9: **Time-sensitivity of CiA: Part 1.** We repeat the experiment shown in Fig 5 (a)/(b) of the main paper for all datasets. Rows represent datasets while columns represent different properties of the task. (a) A single-frame baseline tends to do well on frames at the end of the video sequence since those usually encode either the start or end state of the action. (b) A two-frame baseline usually does best if the frames are picked at the two ends of the video. (c) As more frames are considered in the context, accuracy on chiral action recognition improves. Overall, these demonstrate that chiral action recognition is time-sensitive: it benefits predictably from more frames, especially at the ends.

## C Experiments

### C.1 Setup details

**Details for chiral action recognition.**    To benchmark a given video model for chiral action recognition, we require a single descriptor vector for a video. There are two important details here: (i) *input processing pipeline*: Different methods differ in the way they sample frames, apply cropping operations, etc. (ii) *pooling*: existing methods [5, 19, 79] usually only represent short clips (sequence of frames with a fixed stride), so we need a way of pooling clip-level descriptors into a video-level descriptor. For (i), we follow the data pipeline for each model as provided. For (ii), depending on the method, we either average pool per-clip representations following [43] (*e.g.*, for VideoMAEv2 [87]) or concatenate them (*e.g.*, for 3D ResNet methods like TCLR [19]), or we hand-craft a pooling mechanism (*e.g.*, averaging spatial tokens for each frame and concatenating across time for Intern-Video2.5 [88]). Investigating a general pooling method that gives more time-aware descriptors is an avenue we leave for future work. For image-based model, we sample $T$ frames linearly and simply concatenate per-frame features to represent the video.

**Details for standard action recognition.**    For the experiments with probing video models [40, 79, 89] with LiFT, we sample a single clip of $T=16$ frames with a stride of $s=4$, resize the short side and center crop to $(224, 224)$. Since VideoMAE, VideoJEPA and InternVideo2.5 all produce a sequence of space-time tokens without any global CLS token, we compute the average of all tokens to represent

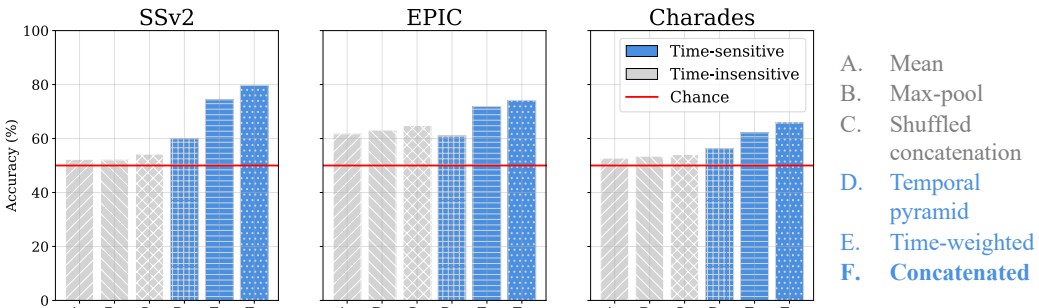

Figure 10: **Time-sensitivity of CiA: Part 2.** We repeat the experiment shown in Fig 5 (c) of the main paper across all three datasets. We show that time-insensitive pooling of per-frame features (*e.g.*, average pooling) leads to much worse performance that with time-sensitive pooling (*e.g.*, concatenation) on chiral action recognition. Note that all the pooling methods considered are non-parametric. This demonstrates the time-sensitivity of chiral action recognition since incorporating the time order of frames substantially improves performance.

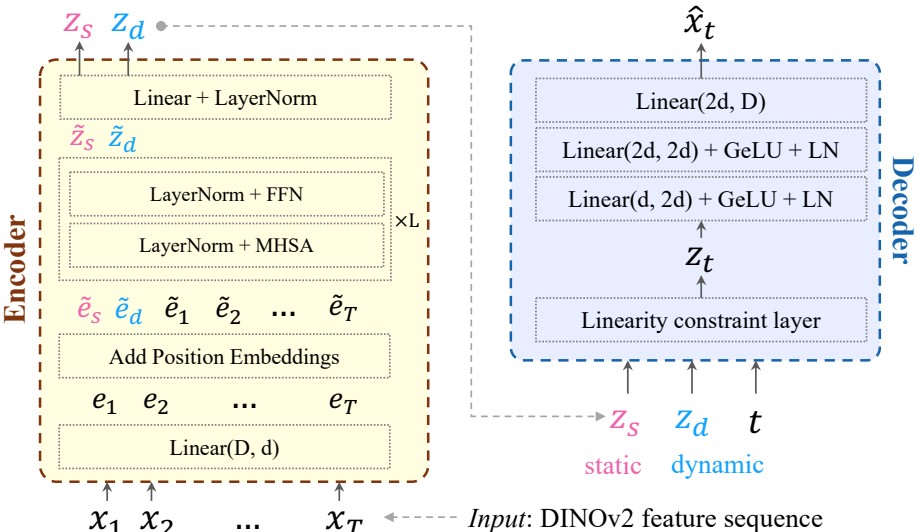

Figure 11: **LiFT architecture details.** The encoder takes in a sequence of DINOv2 features and outputs a video descriptor disentangled into static and dynamic vectors. The decoder reconstructs the feature sequence with linearity baked in the latent space.

the video in case of linear/non-linear probing. Then, we concatenate the LiFT descriptor with this descriptor and train a classifier head on top to output the action class. In case of *linear probe*, the classifier head is a linear layer. In case of *non-linear probing*, it is a two layer MLP with $512$ hidden dimensions with ReLU non-linearity and Dropout of $0.1$. In case of an attentive probe, following [5], we train a single attention layer with a learnable query to pool the space-time tokens into a single descriptor. Then, LiFT is concatenated with the query vector and a linear classifier layer is added on top of the concatenation. We train the probe for 100 epochs using Adam optimizer with learning rate of $1e^{-5}$ and LRPlateau scheduler.

## C.2 Additional Ablations

**Varying the latent dimension** $d$**.** In Tab. 3(a), we vary the latent dimension of the Encoder in LiFT. While the number of parameters increases with $d$, we find that with $d = 384$, LiFT achieves the best performance while still being compact and containing only $8.7M$ parameters. Note that we do not account for the fixed DINOv2 parameters ($22.1M$) in this experiment.

**Varying the amount of training data.** In Tab. Appendix C.2 (b), we vary the amount of training data used to train LiFT with the unsupervised reconstruction loss. We fix the latent dimension to be $d = 384$ and note that the model capacity is fixed. For each row, we run the experiment with three different random seeds and report the average and standard deviation in accuracy. Surprisingly, even with $10\%$ of the data, LiFT gets to 83.3% accuracy. With increasing data samples, the mean accuracy increases marginally. We hypothesize that this is due to two reasons. (i) The model size remains fixed (8.7M) and may not have the capacity to significantly benefit from more samples, (ii) since Kinetics-400 is known to be biased to static understanding (single frame or unordered set of frames [35, 72]), for videos with little visual change, reconstructing the per-frame feature trajectories may not have sufficient signal to inform LiFT. This experiment raises some interesting research questions. Is it possible to achieve time-sensitive video representations by training on selected, *temporally hard* samples only? Is using synthetic data with hand-crafted temporal patterns sufficient [77, 100]? We leave these questions for future work.

| Latent dim $d$ | Accuracy | Parameters (M) |
|---|---|---|
| 192 | 85.9 | 2.3 |
| 256 | 85.8 | 4.0 |
| 384 | **86.6** | 8.7 |
| 512 | 84.9 | 15.3 |

(a) Varying the latent dimension $d$ of Encoder.

| Data frac. | Accuracy |
|---|---|
| 0.1 | $83.3 \pm 0.1$ |
| 0.2 | $84.4 \pm 0.2$ |
| 0.4 | $85.9 \pm 0.4$ |
| 0.6 | $85.6 \pm 0.6$ |
| 0.8 | $85.8 \pm 0.8$ |
| 1.0 | $\mathbf{86.2} \pm 0.4$ |

(b) Varying the $\%$ train data.

Table 8: **Ablations.** Both ablations are conducted on the chiral subset of SSv2. In (a) we vary the latent dimension of the LiFT encoder. We find the best performance with $d = 384$. In (b), we vary the amount of training data (Kinetics-400) used to adapt LiFT. The given $\%$ is uniformly randomly chosen from the entire dataset. Surprisingly, even with $10\%$ of the data, LiFT gets to 83% accuracy. We hypothesize that at fixed model capacity, scaling up to more samples gives diminishing returns.

**Error bars.** To compute error bars, we train LiFT on Kinetics-400 with five different random seeds. The rest of the training configuration is kept constant across all runs. Then, we evaluate the trained models on our main task: chiral action recognition as described in the main paper across the three datasets, SSv2, Charades and EPIC-Kitchens. We report the mean and standard deviation in accuracy in Table 9. The table illustrates these results, highlighting the consistency of the model's performance.

| Dataset | Accuracy (%) |
|---|---|
| SSv2 | $86.1 \pm 0.3$ |
| EPIC | $76.5 \pm 0.8$ |
| Charades | $70.3 \pm 0.6$ |

Table 9: **Error bars for LiFT.** Mean accuracy across five random seeds. LiFT remains fairly stable and the error bars emphasize the difference between LiFT and other video models.

## C.3 Qualitative results

In Fig. 12, we show more examples with tSNE embeddings of the LiFT reconstructed feature trajectories. In most cases, LiFT reconstructs a smoother, continuous approximation of the original trajectory. Note that the original trajectory points seem more scattered than they actually are because tSNE optimizes for local neighborhood distances, which are dominated by closeness of points in the reconstructed trajectory. In case of Fig. 12(f), we observe a divergence between the true and reconstructed trajectories. In this case, the model likely fails to capture the (subtle) visual change which likely causes $\mathbf{z}_d$ to be inaccurate leading to the discrepancy.

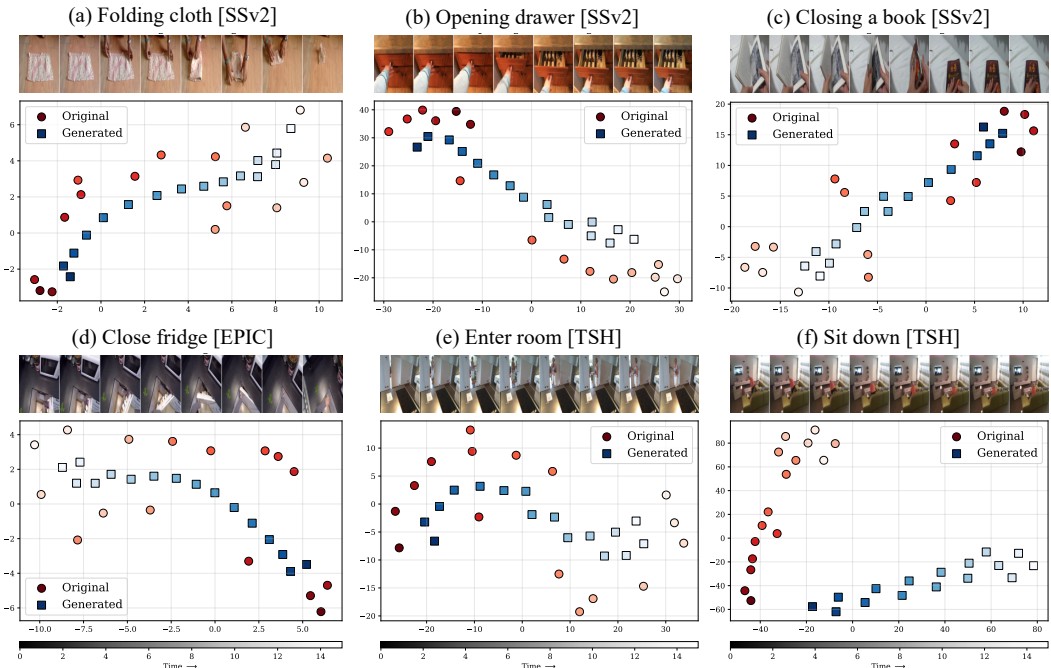

Figure 12: **More qualitative samples of reconstructed features.** We show tSNE embeddings of original and reconstructed features for six videos. The red circles represent original features while blue squares represent reconstructions. Gradient of the color encodes the frame index (time). In general, LiFT tends to output a smooth, continuous approximation of the original feature trajectories in DINO space. (f) is an example failure case where the static token seems reasonable but the direction token is inaccurately predicted causing the direction of original and reconstructed trajectores to differ.

## C.4 The curious case of horizontal motion

Based on Table 4, it seems that correctly encoding horizontal motion (distinguish between something moving → and →) is much harder than encoding vertical motion. Notably, we find that the base model itself (DINOv2-ViT-S/14) struggles with this kind of horizontal motion. Ideally, a change in horizontal spatial position ("moving left to right" vs "moving right to left") should result in dynamic tokens pointing in opposite directions. But this is conditioned on the base model reliably encoding the horizontal spatial position of an object at a given time. Our experiments in Table 10 confirm that the base model itself (DINOv2) does not accurately encode the horizontal spatial position of an object.

| Change type | VideoMAE | VJEPA | DINOv2 (concat.) | LiFT |
|---|---|---|---|---|
| Distance between objects | 70.8 | 87.5 | 83.3 | **87.5** |
| Object count | 64.2 | 62.4 | 69.5 | **72.4** |
| Object size/depth | **96.8** | **96.8** | 92.2 | **96.8** |
| Object state | 72.9 | 66.3 | 75.9 | **80.7** |
| Spatial position ↔ | **96.3** | 96.1 | 75.7 | 75.2 |
| Spatial position ↕ | 91.5 | 89.7 | 79.7 | **93.6** |
| Average | 82.1 | 83.1 | 79.4 | **84.4** |

Table 10: **LiFT is comparable or superior to much larger video models for all types of visual changes except horizontal shift.** On horizontal shift (e.g., "Pulling something from left to right vs. right to left"), LiFT is worse than these video models. As evident from the `DINOv2 (concat.)` column, we confirm that this is because the concatenated base DINOv2 features do not encode such motion as well as the video models.

| Change type | LiFT | (1.) w/ 224 → 448 | (2.) w/ WebSSL | (3.) w/ TTR |
|---|---|---|---|---|
| Distance between objects | 87.5 | **95.8** | 91.7 | 91.7 |
| Object count | 72.4 | 73.7 | **73.9** | 73.2 |
| Object size/depth | 96.8 | 92.9 | 92.2 | **96.9** |
| Object state | **80.7** | 80.3 | **80.7** | 79.2 |
| Spatial position ↔ | 75.2 | **82.4** | 79.7 | 77.9 |
| Spatial position ↕ | 93.6 | **94.4** | 92.8 | 92.4 |
| Average | 84.4 | **86.6** | 85.2 | 85.2 |

Table 11: **We show three directions that improve the performance on encoding horizontal shift motion.** TTR denotes test-time rotation augmentation.

Furthermore, we dug deeper into analyzing the DINOv2 feature sequences for horizontal vs vertical shift samples. We hypothesize that the root cause of the difference in performance of DINOv2 concat. (and consequentially, LiFT) on horizontal vs vertical shifts is due to anisotropic sensitivity of DINO feature sequence, i.e., DINO features vary less with horizontal movement vs vertical movement. Below, we explain our experimental setup and the observations.

To have a perfectly controlled test setting, we generate $N{=}2000$ synthetic sequences with a checkerboard background and a colored disc that moves either horizontally (from left end of the image to right) or vertically (from top to bottom) at a constant rate. An example sequence of horizontal motion is shown in Fig. 13(a). We compute the DINOv2 feature vector for each frame in the sequence. To measure the variation over time, we compute the variance over time and then average it across the feature dimensions. We call this Time Variance (TV). We compute the TV for each sequence and then average it over all sequences for horizontal (or vertical) shifts.

We find that mean Time Variance in vertical shift sequences is about 25% higher than that in horizontal shift sequences. This supports our hypothesis about inherent anisotropic sensitivity of DINOv2 features in case of horizontal or vertical shift motion. Even qualitatively, as shown in Fig. 13(b), vertical motion sequences of DINOv2 concatenated embeddings seem more separable compared to horizontal motion sequences.

It is worth asking why this difference is observed in horizontal vs vertical motion. Is it something to do with the DINO's training procedure (e.g., cropping mechanism) or position encodings in DINO or something else? Likewise, this connects to how we remedy this (e.g., by training DINO with rotated images?). All these questions require more time and deeper investigation and we defer them to future work. However, we do offer three directions that improve the performance on horizontal motion.

**Possible mitigation.** We highlight three promising directions to fix this. In the following, we show the resulting improvements in Table 11, and then explain the rationale for each direction below.

- Scaling up the image resolution at test-time: we hypothesize that encoding of fine-grained information such as the spatial positions of objects should improve with image resolution. This provides a +7.2 point improvement in the spatial position while improving the average across all types of changes by 2.2%.
- Improving the base model: an inherently better model should encode spatial positions better. We use WebSSL [2] which is a scaled up DINO-like image model trained on 2B samples. It yields a boost of +4.5% on horizontal shift.
- Using image rotations as a form of test-time recovery: interestingly, we note that encoding of position along the vertical axis is better than that along the horizontal axis. We exploit this fact and concatenate embeddings of videos rotated by $\pi/2, \pi, 2\pi/3$ with that of the upright video.

There is still a gap of 14% between best LiFT model and VideoMAE on horizontal shift. There is more work to do here but we re-iterate that LiFT is stronger than the video encoders for all other kinds of visual change.

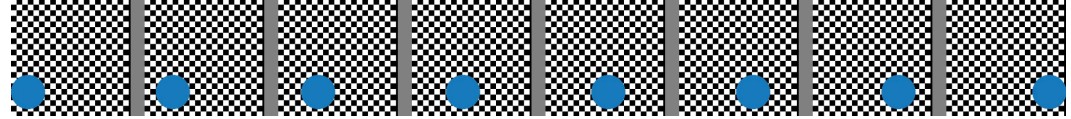

(a) Example synthetic video showing horizontal motion

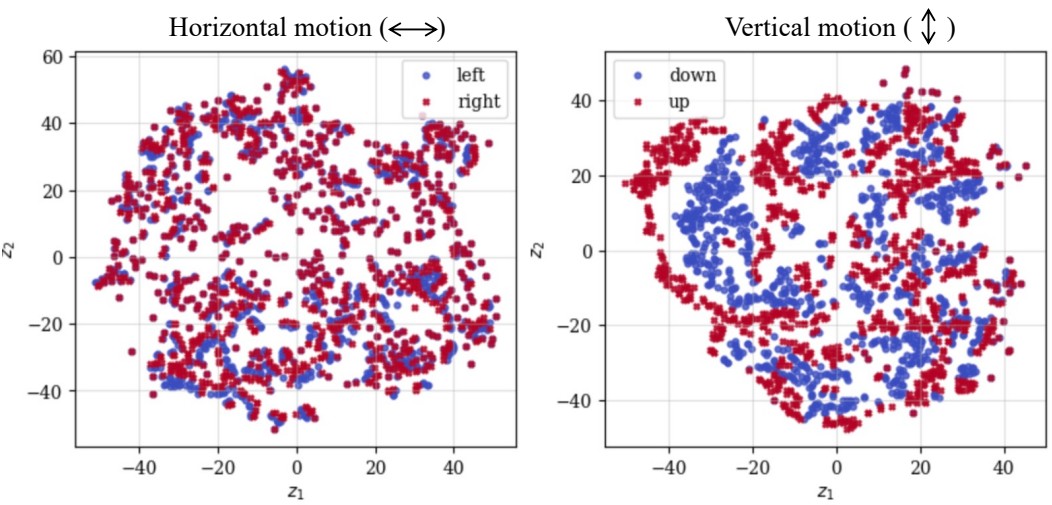

(b) tSNE projections of DINOv2 (concatenated over time) features.

Figure 13: **Synthetic sequence experiments for horizontal vs vertical motion.** Concatenated DINOv2 features are more separable in distinguishing two kinds of vertical motion sequences as compared to horizontal motion sequences.

