# OpenReview forum: "Chirality in Action: Time-Aware Video Representation Learning by Latent Straightening"
_NeurIPS.cc/2025/Conference — NeurIPS 2025 poster_

### Official Review · Reviewer_FvAQ · 2025-07-03

**Clarity:** 4
**Significance:** 3
**Originality:** 3
**Rating:** 5
**Confidence:** 4

**Summary:**

This paper focuses on learning time-sensitive video representations. To achieve this, it (1) introduces a new task, chiral action recognition, where the model distinguishes between temporally opposite actions, and (2) proposes Linearized Feature Trajectories (LiFT), a method that injects time-sensitivity into frozen DINOv2 features. LiFT enforces linearity in the latent space while ensuring orthogonality between static and dynamic vectors. Experimental results demonstrate significant improvements in the proposed chiral action recognition task, with reasonable gains in standard action recognition tasks.

**Questions:**

Please see Weaknesses section.

**Ethical Concerns:**

["NO or VERY MINOR ethics concerns only"]

**Final Justification:**

My initial concerns centered on the latent space assumption, potential degradation of spatial features, and missing related literature. All of these concerns have been thoroughly addressed during the rebuttal and discussion phase. Overall, I find this
paper original and interesting, with sufficient merit for publication to impact the video understanding community. While minor weaknesses remain, such as the lack of comparison with 3D video models (as raised by reviewer VxBp), I believe the current paper
meets the standards for acceptance. Therefore, I recommend 5: Accept.

**Limitations:**

yes

**Quality:**

3

**Strengths And Weaknesses:**

### Strengths

1. **Interesting Hypothesis:** The paper suggests an interesting idea for building video representations—per-frame features form a smooth trajectory in latent space, making the representation time-sensitive. This concept is inspired by human perception theory and is worth exploring.
2. **Originality:** Many have tried to separate static and dynamic information, but this paper offers a fresh approach to solving that problem.
3. **Clear Presentation:** The paper is well-written, easy to follow, and clearly explains the concepts, making them simple to understand.

### Weaknesses

Overall, I like the concept and direction of this paper. It presents a refreshing approach. However, there are a few concerns that need to be addressed before we can conclude that the proposed method effectively learns good video features:

1. **Latent Space Assumption:** As stated in Equation 4, this method assumes a constant difference between consecutive frames in the learned latent space. However, real-world actions do not typically occur at a perfectly constant speed. Forcing a constant step size in the latent space could potentially limit the model's ability to accurately represent actions with highly variable speeds or sudden changes. Given that the paper primarily compares performance on short, relatively uniform videos, it would be valuable to see experimental results with more dynamic video benchmarks, such as long video recognition benchmark, to address this concern.
2. **Potential Degradation on Spatial Features:** The linear step size in the learned latent space is a strong constraint, and I'm concerned that the improvements in handling temporal variations might come at the cost of losing spatial details. For instance, Table 4 shows that the method performs well for several attributes but struggles with the "Spatial position ↔" change. To investigate this, it would be helpful for the authors to include the "DINOv2 ViT-S/14 Concat." baseline in Table 4. This would clarify whether LiFT's lower performance on this type of change is due to a degradation or an inherent limitation of the base features.
3. **Missing Relevant Dataset:** The “action antonym” tasks in the TVBench dataset serve the same purpose as the CiA dataset—distinguishing between two exactly opposite actions. It would be helpful to include a discussion of this overlap in the related work and highlight the differences between the datasets.[TVBench] Cores, Daniel, et al. "TVBench: Redesigning Video-Language Evaluation." *arXiv preprint arXiv:2410.07752* (2024).

---

> ### Author Rebuttal · Authors · 2025-07-30
>
> Thank you for the encouraging feedback. We are glad to see that you find the idea interesting, novel and clearly presented. We address your concerns below.
>
> ---
>
> ### Latent space assumption
>
> > As stated in Equation 4, this method assumes a constant difference between consecutive frames in the learned latent space. However, real-world actions do not typically occur at a perfectly constant speed. Forcing a constant step size in the latent space could potentially limit the model's ability to accurately represent actions with highly variable speeds or sudden changes.
>
> We emphasize that the linearity is imposed only in the latent space of the autoencoder that takes in the sequence of DINOv2 features. The trajectory of change in the pixel space or in the DINOv2 space can be non-linear but is mapped to a linearised latent space. This is also evident in qualitative visualization of reconstructed features trajectories in Fig C.2. in the supplementary material. LiFT is also able to reconstruct trajectories that are non-linear in the DINO space.
>
> > Given that the paper primarily compares performance on short, relatively uniform videos, it would be valuable to see experimental results with more dynamic video benchmarks, such as long video recognition benchmark, to address this concern.
>
>  Our objective is to get a compact time-sensitive representation of a short video that encodes the time-varying parts in the video beyond just the static parts. To take on more complex tasks, e.g., long videos that involve multiple events, we believe that we first need strong temporal representations for each event, which is where our work excels.
>
> ---
>
> ### Potential Degradation on Spatial Features
>
> Thank you for the suggestion. This also co-incides with the points raised by Reviewer `hKBo`. We respond with a common set of experiments repeated here for easier readability.
>
> **Diagnosing spatial position ↔.** We did conduct an experiment with the "DINOv2 ViT-S/14 Concat." compared to LiFT with different kinds of visual change. The results are reported in *Table T1* below. Indeed, we confirm that the underperformance of LiFT on understanding horizontal shift (`Spatial position ↔ `) boils down to the base model not encoding the spatial position along the $x$-axis reliably. The results are shown below in *Table T1*.
>
>
> | Change type              | VideoMAE | VJEPA | DINOv2 (concat.) | LiFT |
> |--------------------------|:-----------:|:-------:|:------------------:|:----------------:|
> | Distance between objects | 70.8      | 87.5  | 83.3             | **87.5**           |
> | Object count             | 64.2      | 62.4  | 69.5             | **72.4**           |
> | Object size/depth        | 96.8      | 96.8  | 92.2             | **96.8**           |
> | Object state             | 72.9      | 66.3  | 75.9             | **80.7**           |
> | Spatial position ↔       | **96.3**      | 96.1  | 75.7             | 75.2           |
> | Spatial position ↕       | 91.5      | 89.7  | 79.7             | **93.6**           |
> | Average              | 82.1      | 83.1  | 79.4             | **84.4**           |
>
> *Table T1.* LiFT is comparable or superior to much larger video models for all types of visual changes except horizontal shift (e.g., "Pulling something from left to right vs. right to left"). On horizontal shift, LiFT is substantially worse than these models. As evident from the `DINOv2 (concat.)` column, we confirm that this is because the concatenated base DINOv2 features do not encode such motion as well as the video models.
>
>
> **Mitigation:** We highlight three directions which show promise in mitigating this issue. The results are shown below in *Table T2*.
>
> | Change type              | LiFT  | (1.) w/ 224 → 448 | (2.) w/ WebSSL | (3.) w/ test-time rotations |
> |--------------------------|:-------:|:---------------:|:-----------:|:------------------------:|
> | Distance between objects | 87.5  | **95.8**          | 91.7      | 91.7                   |
> | Object count             | 72.4  | 73.7          | **73.9**      | 73.2                   |
> | Object size/depth        | 96.8  | 92.9          | 92.2      | **96.9**                   |
> | Object state             | 80.7  | 80.3          | **80.7**      | 79.2                   |
> | Spatial position ↔       | 75.2  | **82.4**          | 79.7      | 77.9                   |
> | Spatial position ↕       | 93.6  | **94.4**          | 92.8      | 92.4                   |
> | Average             | 84.4  | **86.6**          | 85.2      | 85.2                   |
>
> *Table T2.* We show three directions that improve the performance on encoding horizontal shift motion.
>
> 1. *Scaling up the image resolution at test-time*: we hypothesize that encoding of fine-grained information such as the spatial positions of objects should improve with image resolution following Shi et al [1]. This provides a +7.2 point improvement in the spatial position while improving the average across all types of changes by 2.2%.
> 2. *Improving the base model*: an inherently better model should encode spatial positions better. We use WebSSL [2] which is a scaled up DINO-like image model trained on 2B samples. It yields a boost of +4.5% on horizontal shift.
> 3. *Using image rotations as a form of test-time recovery*: interestingly, we note that encoding of position along the vertical axis is better than that along the horizontal axis. We exploit this fact and concatenate embeddings of videos rotated by $[\pi/2, \pi, 2\pi/3]$ with the upright video.
>
> There is still a gap of ~14% between best LiFT model and VideoMAE on horizontal shift. We acknowledge that there is more work to do but re-iterate that LiFT is stronger than the video encoders for all other kinds of visual change.
>
> [1] When Do We Not Need Larger Vision Models? Baifeng Shi, Ziyang Wu, Maolin Mao, Xin Wang, Trevor Darrell. arXiv:2403.13043
>
> [2] Scaling Language-Free Visual Representation Learning. Fan et al. arXiv:2504.01017
>
> ---
>
> ### Related dataset
>
> > Missing Relevant Dataset: : The “action antonym” tasks in the TVBench dataset serve the same purpose as the CiA dataset—distinguishing between two exactly opposite actions. It would be helpful to include a discussion of this overlap in the related work and highlight the differences between the datasets.[TVBench] Cores, Daniel, et al. "TVBench: Redesigning Video-Language Evaluation." arXiv preprint arXiv:2410.07752 (2024).
>
> Thank you for pointing us to this paper – this is indeed relevant. We highlight some differences:
>
>  (i) TVBench is based on only 1 dataset (NTU) with 320 videos across 8 antonym pairs. Ours is based on 3 datasets with 62,363 videos across 110 chiral pairs.
>
> (ii) TVBench is designed for video-text models while we focus on video embeddings only. It is our plan to extend our benchmark with text annotations in follow-up work.
>
> We will include this discussion in the final version and perhaps expand a version of our benchmark by including their samples.

---

> ### Comment · Reviewer_FvAQ · 2025-08-04
>
> Thank you to the authors for providing a detailed rebuttal. My concerns regarding Q1 and Q3 are adequately resolved. Regarding Q2, I have one follow-up question. Based on the additional results in Table T1 and Table T2, it's evident that LiFT shows advantages for all other change types except for horizontal shift (Spatial position ↔). Is there a rationale or assumption that the authors believe explains this? I think "this is because the concatenated base DINOv2 features do not encode such motion as well as the video models" is not an enough explanation, as LiFT shows advantages on vertical shifts (Spatial position ↕), i.e., DINOv2 (concat.)	79.7 $\rightarrow$ LiFT 93.6.  I acknowledge that LiFT still outperforms all other change types in general, but I think this would be a valuable point to discuss.

---

> ### Author Response · Authors · 2025-08-07
>
> Thank you again for your constructive and careful feedback. In order to address your follow-up question, we dug deeper into analyzing the DINOv2 feature sequences for horizontal vs vertical shift samples. We hypothesize that the root cause of the difference in performance of DINOv2 concat. (and consequentially, LiFT) on horizontal vs vertical shifts is due to anisotropic sensitivity of DINO feature sequence, i.e., DINO features vary less with horizontal movement vs vertical movement. Below, we explain our experimental setup and the observations.
>
> **Data and metric**. To have a perfectly controlled test setting, we generate $N{=}2000$ synthetic sequences with a checkerboard background and a colored disc that moves either horizontally (from left end of the image to right) or vertically (from top to bottom) at a constant rate. We compute the DINOv2 feature vector for each frame in the sequence. To measure the variation over time, we compute the variance over time and then average it across the feature dimensions. We call this Time Variance (TV). We compute the TV for each sequence and then average it over all sequences for horizontal (or vertical) shifts.
>
> **Observation**. We find that mean Time Variance in vertical shift sequences is about 25% higher than that in horizontal shift sequences. This supports our hypothesis about inherent anisotropic sensitivity of DINOv2 features in case of horizontal or vertical shift motion. We will include this analysis on synthetic sequences along with qualitative tSNE visualizations in the supplementary material of the final paper.
>
> **What causes this? And how to fix it?** The suggestion from the reviewer and our analysis has led to the intriguing question of why DINO has this anisotropic sensitivity to vertical vs horizontal shift motions. Is it something to do with the DINO's training procedure (e.g., cropping mechanism) or position encodings in DINO or something else? Likewise, this connects to how we remedy this (e.g., by training DINO with rotated images?). All these questions require more time and deeper investigation and we defer them to future work.
>
> We thank the reviewer again for raising an interesting observation and their time and effort in reviewing our work.

---

> > ### Comment · Reviewer_FvAQ · 2025-08-07
> >
> > Thank you for your time and effort on this additional experiment! The 25% higher Time Variance (TV) observed in vertical shift sequences compared to horizontal shifts within latent space directly clarifies why LiFT showed limited improvement on "Spatial position ↔" changes. Given that LiFT is designed to capture temporal sensitivity, it's understandable that combining DINO v2 concat with LiFT didn’t significantly enhance performance in horizontal changes. This quantitative analysis and accompanying discussion are very helpful in understanding the results and which way we can improve, and I believe they are worth including in the final version.
> >
> > Overall, the analysis of time variance and Tables T1 & T2 effectively addressed my initial concerns about Q2: whether LiFT's limited gains were due to feature degradation or an inherent limitation of the base features. With all my concerns resolved, I am inclined to raise my score from 4 to 5. I believe this paper provides a fresh direction for the video representation learning community.

---

### Official Review · Reviewer_VxBp · 2025-07-03

**Clarity:** 3
**Significance:** 2
**Originality:** 3
**Rating:** 4
**Confidence:** 4

**Summary:**

The paper introduces chiral action recognition, a novel task that evaluates a video representation's sensitivity to time by distinguishing between temporally opposite actions (e.g., opening vs. closing). To address this, the authors propose LiFT (Linearized Feature Trajectories)—a self-supervised method that transforms per-frame DINOv2 features into smooth linear, time aware trajectories in latent space. LiFT outperforms much larger video models on three chiral datasets and enhances standard action recognition benchmarks when combined with existing models.

**Questions:**

- Have you considered training the image (or video) encoder and LiFT together? Would that improve performance even further?
- In Table 5, the benefits brought by LiFT seem to diminish with the attention probe. Do you have any intuition on the reason?

**Ethical Concerns:**

["NO or VERY MINOR ethics concerns only"]

**Final Justification:**

See my comments to the rebuttal.

**Limitations:**

See weaknesses.

**Quality:**

2

**Strengths And Weaknesses:**

Strengths:
- The authors constructed a chiral action classification benchmark, which are useful for a broader community.
- LiFT + video models have a great boost in action classification performance.

Weaknesses:
- An ideal video representation for action classification should work for both chiral actions and spatial information dominant actions. From Table 2 it’s unclear whether LiFT gets better chiral classification at the cost of other action types.
- In Table 2, comparing with video models + average pooling is not fair, because if an action is reverse played (open VS close door) then these models would get almost the same outputs after average pooling. Why not concatenate them as well?
- Fundamentally I’m curious why the authors chose an image-based model for this task. Theoretically, video-level embedding with 3D convs or transformers should always outperform image models since they can always be reduced to an image model. Since this is an application paper, why do we spend time focusing on the model that’s not designed for the task (action classification)? This model seems to suggest that motion information should be processed (by the small Transformer encoder) after frame-wise features are fully extracted (by DINO), which is hard to be true.
- The paper mentioned “supplement” multiple times but I didn’t seem to find it?

---

> ### Author Rebuttal · Authors · 2025-07-30
>
> Thank you for a thoughtful and constructive review. We appreciate that you find the CiA benchmark useful. Below, we address your concerns.
>
> ---
>
> ### Trading off spatial understanding for temporal understanding?
>
> > An ideal video representation for action classification should work for both chiral actions and spatial information dominant actions. From Table 2 it’s unclear whether LiFT gets better chiral classification at the cost of other action types.
>
> Our intuition for LiFT is that it shall draw its spatial capabilities from a strong image encoder (e.g., DINOv2) and its temporal capabilities from the proposed self-supervised adaptation recipe. We establish superior performance on chiral actions but it is fair to ask if that comes at the cost of spatially-dominant actions.
>
> It depends on what constitutes a spatial information dominant action. If it includes actions that can be recognized just by the spatial context rather than temporal context, then UCF101 and HMDB51 are known to be spatially-biased datasets [1, 2]. In Table 5 of the main paper, we report the performance of LiFT on these datasets in comparison to bigger video models.
>
> Here, we report a more exhaustive set of experiments below in *Table T3* to test possible degradation in understanding spatial actions and possible mitigations. Moreover, we also report performance on `Noun` and `Verb` prediction on EPIC-Kitchens in the same *Table T3*.
>
> | Model         | UCF101   | HMDB51  | EPIC - Nouns | EPIC - Verbs |
> |---------------|:----------:|:---------:|:---------------:|:---------------:|
> | Chance        | 1.0      | 2.0     | 0.4           | 1.0           |
> | DINOv2 (avg.) | 84.7     | 64.3    | 19.4          | 31.5          |
> | DINOv2 (cat.) | 76.1     | 53.3    | 17.6          | 37.4          |
> | VideoMAE      | 83.6     | 66.5    | 22.3          | 47.2          |
> | LiFT (DINOv2) | 86.6     | 65.2    | 18.8          | 40.8          |
> | LiFT (WebSSL) | 89.6     | 74.4    | 22.6          | 44.0          |
> | VideoJEPA     | 91.3 | 76.1 | 24.9      | 47.9      |
> | LiFT (WebSSL) $\oplus$ VideoJEPA |  **94.0** | **78.3** | **26.7** | **50.8** |
>
> *Table T3.*
> (i) On spatially-dominant datasets, LiFT outperforms both naive baselines of mean pooling and concatenation.
> (ii) While bigger video models outperform LiFT on these datasets, using a better base image model (`WebSSL`) strengthens the spatial information in LiFT. This achieves competitive results on spatial actions while outperforming these heavier video models on chiral actions.
> (iii) Combining `LiFT (WebSSL)` features with `VideoJEPA` achieves the best results. This reinforces our main findings in the paper:   `LiFT` is very strong at chiral action recognition and its combination with an existing model is very strong at spatially dominant action recognition. Finally, this again highlights that LiFT is complementary to standard video models.
>
> [1] What Makes a Video a Video: Analyzing Temporal Information in Video Understanding Models and Datasets. Huang et al. CVPR 2018.
>
> [2] Revisiting the "Video" in Video-Language Understanding. Buch et al. CVPR 2022.
>
> ---
>
> ### Experimental setup
>
> > In Table 2, comparing with video models + average pooling is not fair, because if an action is reverse played (open VS close door) then these models would get almost the same outputs after average pooling. Why not concatenate them as well?
>
> 1. These video models do not have a CLS token to represent the video with a single embedding.
> 2. It is not entirely true that average pooling tokens from the last layer will yield the same result even if the action is played in reverse – because video models act on 3D space-time tokens enhanced by temporal position encoding; different frames interact with each other in every layer of the transformer. This is also shown empirically where VideoMAE, despite average pooling, does significantly better than chance on chiral tasks. But average pooling image features (DINOv2) is at chance level because the frames have never interacted with each other and average pooling makes it time-invariant.
>
> Nevertheless, we do repeat experiments where we spatially pool tokens corresponding to each frame and then concatenate tokens over time. The results are shown below in *Table T4*. LiFT still outperforms these variants.
>
> | Model           | Pooling      | Embedding size ↓            | Chiral Accuracy (avg.) ↑         |
> |----------------|--------------|:------------------------------:|:----------------------------------:|
> | Chance          | -            | -                            | 50.0                             |
> | VideoMAE        | Average      | 1024                         | 70.3                             |
> | VideoMAE        | Time concat. | 8192                         | 75.6                             |
> | VideoJEPA       | Average      | 1024                         | 68.1                             |
> | VideoJEPA       | Time concat. | 8192                         | 71.1                             |
> | InternVideo 2.5 | Average      | 4096                         | 59.1                             |
> | InternVideo 2.5 | Time concat. | 32768                        | 71.2                             |
> | LiFT    | -            | **768**                          | **78.3**                         |
>
> *Table T4.* Results with concatenating video tokens along time still underperforms LiFT. The reported accuracy is average across the three datasets in CiA.
>
>
> ---
>
> ### Modelling choice
>
> > Fundamentally I’m curious why the authors chose an image-based model for this task. Theoretically, video-level embedding with 3D convs or transformers should always outperform image models since they can always be reduced to an image model. Since this is an application paper, why do we spend time focusing on the model that’s not designed for the task (action classification)?
>
> While an image model (DINOv2) is not designed for action recognition, it has multiple benefits:
> - Given the massive scale of image training data, it already encodes the static information very well. Our intuition behind LiFT is that we rely on DINO for static spatial information and only "add" temporal knowledge on top of it with self-supervision.
> - It is computationally much more efficient to adapt an image model than training pure video models from scratch.
>
> > This model seems to suggest that motion information should be processed (by the small Transformer encoder) after frame-wise features are fully extracted (by DINO), which is hard to be true.
>
> One of the key findings in our paper is that trajectories of per-frame features  over time still capture a surprisingly large amount of high-level temporal information, e.g. how the object-state is changing over time,  which LiFT extracts well as shown. But for something very low-level, e.g., tracking the motion of a small object in a video, perhaps it is better to use pure video models as you suggest.
>
> ---
>
> ### Misc.
>
> > Have you considered training the image (or video) encoder and LiFT together? Would that improve performance even further?
>
> This is an excellent suggestion. Due to limited computational resources and time, we did not experiment with training from scratch. However, we are equally curious to see if the LiFT self-supervised recipe can be trained jointly with standard video recipes such as masked modelling with VideoMAE. We leave this for future work.
>
> > In Table 5, the benefits brought by LiFT seem to diminish with the attention probe. Do you have any intuition on the reason?
>
> This is a good observation. We think this is due to two reasons: (i) Attentive probes have a surprisingly large number of parameters giving them more capacity (12.5M with 1024-D embeddings), (ii) since attentive probe operates on all space-time tokens, it already enables some level of temporal modelling so the benefit of combining with LiFT is relatively lower. Having said that, we do note there is still a noticeable benefit (+1 point with VJEPA and +2.2 with VideoMAE).
>
> > The paper mentioned “supplement” multiple times but I didn’t seem to find it?
>
> The supplementary material was submitted as a `.zip` in OpenReview. It includes a PDF and the CiA dataset metadata.

---

> > ### Comment · Reviewer_VxBp · 2025-08-03
> >
> > I appreciate the thorough response given by the author. The evidence of LiFT not hurting spatial-information-dominated action classes is very encouraging. The fact that LiFT outperforms video models with frame-concatenated latents adds great value to the comparisons. That being said, I wish the authors can explore more on the practical aspect of LiFT -- for example, it might be computationally much cheaper than a full 3D video model. I agree that the observation that per-frame trajectory over time carries surprisingly rich information is valuable to the community. Therefore, I will increase my score from 3 to 4.

---

> > > ### Author Response · Authors · 2025-08-03
> > > **Appreciate the encouraging feedback**
> > >
> > > We thank the Reviewer for their positive and constructive engagement with our work.
> > >
> > > Indeed, we are also encouraged by LiFT's strong performance on spatial-information-dominant actions along with the temporal chiral actions.
> > >
> > > We agree that the practical aspects of LiFT should be explored further. In the final version, we will include a discussion on aspects like the compute cost, context length, size of training datasets, etc. of LiFT compared to 3D video models.

---

### Official Review · Reviewer_hKBo · 2025-07-03

**Clarity:** 2
**Significance:** 2
**Originality:** 2
**Rating:** 3
**Confidence:** 4

**Summary:**

Current video foundation models’ representations still get confused between actions which look similar but are time-sensitive, such as opening a door vs closing a door, folding a paper vs unfolding a paper, moving from left to right and right to left. These actions defined by the authors are called Chiral Action Pairs. This paper introduces Linearized Feature Trajectory (LiFT) which aims to distinguish such chiral actions, which are time-sensitive, such as opening a door and closing a door, etc. The authors achieve this by constructing time-aware representation that allows linear separability of chiral pairs by identifying two summary tokens z_s (static) and z_d (dynamic). The authors benchmark the efficacy of their algorithm on the Chirality in Action (CiA) dataset, developed in this research by identifying and clubbing different chiral groups in three datasets, such as SSV2, EPIC and Charades. Benchmark results show that their method surpasses existing video foundation models on such chiral action recognition and improves existing benchmarks as well when combined with existing methods.

**Questions:**

- Could you please diagnose where the spatial bias of LiFT is coming from, as shown in Table 4? Is it really from the global pooling or some augmentation techniques used? For example, you could disable flips in your experiments. The drop looks significant, and I’d request authors to provide appropriate solution to prevent such degradation.

- The evaluation protocol is still simple. Could you please report results on at least one task that needs multi-step reasoning?

**Ethical Concerns:**

["NO or VERY MINOR ethics concerns only"]

**Limitations:**

While the authors mention the limitation at the end of the paper, they do not mention the societal impacts in the paper. Even if the authors feel that there is no negative societal impact (as mentioned in the NeurIPS checklist), they ignore positive societal impact. It would be better to write about societal impacts in general.

**Quality:**

3

**Strengths And Weaknesses:**

Strengths

•	The paper introduces a fresh test chiral action recognitions that really checks whether a model understands the direction of an action instead of just spotting static background clues. Building a meta-dataset from three well-known video sets makes the benchmark easy for others to adopt.

•	LiFT is basically an “adapter” that could be plugged on top of the popular DINO-v2 image encoder. After self-supervised training, it turns every video into just two compact vectors z_s and z_d, that clearly separate “what stays” from “what moves”, respectively.

•	Despite being tiny, LiFT beats much larger video transformers, models ten times bigger and trained on six times more clips by over 7% on the new chiral benchmark. That shows the design targets the right signal (temporal change).

•	LiFT is complementary rather than redundant as concatenating LiFT’s descriptors with those from standard models improves the performance, as shown in Table 5.

Weaknesses

•	The evaluation reduces each inverse-action pair to an isolated, binary, linear separability test, it rewards embeddings that encode a single shortcut signal while overlooking multi-step dynamics. As a result, a model can excel on the benchmark proposed in the paper and yet fail to generalize to realistic tasks that require long-range temporal reasoning, compositional actions, or non-linear decision boundaries.

•	LiFT collapses each frame into a global feature and then forces the whole video to follow a straight-line “time path,” which tends to keep track of what changes but washes out where things are. As a result, it struggles to tell “object on the left” from “object on the right,” scoring far below competing models on left-versus-right tasks, as also evident in Table 4, where compared to VMAE and VJEPA, LiFT performance is greatly degraded.

---

> ### Author Rebuttal · Authors · 2025-07-29
>
> Thank you very much for the detailed and constructive feedback. Below, we address your concerns part by part.
>
> ---
>
> ### Loss in spatial position information
>
> > LiFT collapses each frame into a global feature and then forces the whole video to follow a straight-line “time path,” which tends to keep track of what changes but washes out where things are. As a result, it struggles to tell “object on the left” from “object on the right,” scoring far below competing models on left-versus-right tasks, as also evident in Table 4, where compared to VMAE and VJEPA, LiFT performance is greatly degraded.
>
> > Could you please diagnose where the spatial bias of LiFT is coming from, as shown in Table 4? Is it really from the global pooling or some augmentation techniques used? For example, you could disable flips in your experiments. The drop looks significant, and I’d request authors to provide appropriate solution to prevent such degradation.
>
>
> First, we clarify that LiFT aims to learn both: *what is changing* (through the static token) and *how is it changing* (through the dynamic token). In a crude analogy, the former would be a point on the unit sphere in $\mathbb{R}^{d}$ encoding the spatial context and the latter a tangent (direction of change). Since there is a static token, LiFT does not "wash out where things are".
>
>
> **Diagnosis of spatial bias**: It is certainly due to the base image model (DINOv2 ViT-S/14) not encoding this horizontal spatial information sufficiently, as explained below. We do not use any global pooling or horizontal flips during training or evaluation.
>
> Ideally, a change in horizontal spatial position (“moving left to right” vs “moving right to left”) should result in dynamic tokens pointing in opposite directions. But this is conditioned on the base model reliably encoding the horizontal spatial position of an object at a given time. Our experiments in *Table T1* confirm that the base model itself (DINOv2)  does not accurately encode the horizontal spatial position of an object.
>
>
> | Change type              | VideoMAE | VJEPA | DINOv2 (concat.) | LiFT |
> |--------------------------|:-----------:|:-------:|:------------------:|:----------------:|
> | Distance between objects | 70.8      | 87.5  | 83.3             | **87.5**           |
> | Object count             | 64.2      | 62.4  | 69.5             | **72.4**           |
> | Object size/depth        | 96.8      | 96.8  | 92.2             | **96.8**           |
> | Object state             | 72.9      | 66.3  | 75.9             | **80.7**           |
> | Spatial position ↔       | **96.3**      | 96.1  | 75.7             | 75.2           |
> | Spatial position ↕       | 91.5      | 89.7  | 79.7             | **93.6**           |
> | Average              | 82.1      | 83.1  | 79.4             | **84.4**           |
>
> *Table T1.* LiFT is comparable or superior to much larger video models for all types of visual changes except horizontal shift. On horizontal shift (e.g., "Pulling something from left to right vs. right to left"), LiFT is worse than these video models. As evident from the `DINOv2 (concat.)` column, we confirm that this is because the concatenated base DINOv2 features do not encode such motion as well as the video models.
>
> **Possible mitigation**: The reviewer raises an interesting point on “appropriate solution to prevent such degradation”. We conduct additional experiments and highlight three promising directions to fix this. In the following, we show the resulting improvements in *Table T2*, and then explain the rationale for each direction below the table.
>
> | Change type              | LiFT  | (1.) w/ 224 → 448 | (2.) w/ WebSSL | (3.) w/ test-time rotations |
> |--------------------------|:-------:|:---------------:|:-----------:|:------------------------:|
> | Distance between objects | 87.5  | **95.8**          | 91.7      | 91.7                   |
> | Object count             | 72.4  | 73.7          | **73.9**      | 73.2                   |
> | Object size/depth        | 96.8  | 92.9          | 92.2      | **96.9**                   |
> | Object state             | 80.7  | 80.3          | **80.7**      | 79.2                   |
> | Spatial position ↔       | 75.2  | **82.4**          | 79.7      | 77.9                   |
> | Spatial position ↕       | 93.6  | **94.4**          | 92.8      | 92.4                   |
> | Average             | 84.4  | **86.6**          | 85.2      | 85.2                   |
>
> *Table T2.* We show three directions that improve the performance on encoding horizontal shift motion.
>
> 1. *Scaling up the image resolution at test-time*: we hypothesize that encoding of fine-grained information such as the spatial positions of objects should improve with image resolution following Shi et al [1]. This provides a +7.2 point improvement in the spatial position while improving the average across all types of changes by 2.2%.
> 2. *Improving the base model*: an inherently better model should encode spatial positions better. We use WebSSL [2] which is a scaled up DINO-like image model trained on 2B samples. It yields a boost of +4.5% on horizontal shift.
> 3. *Using image rotations as a form of test-time recovery*: interestingly, we note that encoding of position along the vertical axis is better than that along the horizontal axis. We exploit this fact and concatenate embeddings of videos rotated by $[\pi/2, \pi, 2\pi/3]$ with the upright video.
>
> There is still a gap of ~14% between best LiFT model and VideoMAE on horizontal shift. We acknowledge that there is more work to do but re-iterate that LiFT still remains very strong in encoding all other kinds of visual change - stronger than the video encoders for all other kinds of visual change.
>
> [1] When Do We Not Need Larger Vision Models? Baifeng Shi, Ziyang Wu, Maolin Mao, Xin Wang, Trevor Darrell. arXiv:2403.13043
>
> [2] Scaling Language-Free Visual Representation Learning. Fan et al. arXiv:2504.01017
>
> ---
>
> ### Evaluation protocol
>
> > The evaluation reduces each inverse-action pair to an isolated, binary, linear separability test, it rewards embeddings that encode a single shortcut signal while overlooking multi-step dynamics.
>
> We conduct a linear separability test for each pair separately (e.g., folding/unfolding paper) because we want to **systematically test if a video embedding encodes the particular visual change** necessary to distinguish between the given temporally opposite actions (e.g., size of the paper). We use a light linear probe on *frozen* embeddings to avoid fitting to specific shortcut signals and rely on the desired temporal signal.
>
>
> > As a result, a model can excel on the benchmark proposed in the paper and yet fail to generalize to realistic tasks that require long-range temporal reasoning, compositional actions, or non-linear decision boundaries.
>
> 1. Our benchmark (CiA) is designed to measure one desirable aspect of a video embedding: time sensitivity to visual change. It is meant to complement existing benchmarks that tend to focus on spatial information.
> 2. We do not claim to have a general model that can do long-range temporal reasoning, understand compositional actions, etc. Our objective is to get a compact time-sensitive representation of a short video that encodes the time-varying parts in the video beyond just the static parts. To take on more complex tasks involving multiple events, we first need strong temporal representations for the video clip of each event, which is where our work excels.
> 3. The model not only excels on the proposed benchmark,  it is also able to boost the performance on standard benchmarks by combining it with standard models (as shown in Table 5). Thus, there is no evidence that it will not 'generalize to realistic tasks that require long-range temporal reasoning, compositional actions, or non-linear decision boundaries', quite the opposite in fact.
>
>
> **Societal impact**
>
> > While the authors mention the limitation at the end of the paper, they do not mention the societal impacts in the paper. Even if the authors feel that there is no negative societal impact (as mentioned in the NeurIPS checklist), they ignore positive societal impact. It would be better to write about societal impacts in general.
>
> Thank you for this suggestion. We will include a discussion on the societal impacts in general, and particularly in light of domestic robots that will encounter a lot of these everyday chiral actions such as opening/closing a door, turning on/off light, etc.

---

> > ### Comment · Area_Chair_xdRL · 2025-08-05
> >
> > Dear Reviewer, you are the one reject review. The authors have left detailed comments, please read them and ask any needed clarifying questions. Also read the other reviews and engage in discussion with those reviewers to come to a consensus for this paper.

---

### Official Review · Reviewer_1AZN · 2025-07-03

**Clarity:** 3
**Significance:** 3
**Originality:** 3
**Rating:** 5
**Confidence:** 4

**Summary:**

This paper presents a novel approach to time-aware video representation learning through a method called LiFT (Linearized Feature Trajectories). The central idea is that per-frame features of a video lie on smooth trajectories that encode temporal change, and these can be mapped into a latent space where they become approximately linear. This is inspired by a human vision study on Perceptual Straightening Hypothesis. The authors propose a lightweight, self-supervised model that adapts frozen image features (from DINOv2) using a Transformer encoder and a latent linearity constraint.
To evaluate time-sensitivity, the authors introduce a new benchmark task—chiral action recognition—and construct a meta-dataset (CiA) from existing video datasets (SSv2, EPIC-Kitchens, Charades), focusing on pairs of temporally opposite actions (e.g., opening vs. closing). The evaluation protocol uses linear probes to assess the separability of these chiral pairs.

**Questions:**

--

**Ethical Concerns:**

["NO or VERY MINOR ethics concerns only"]

**Final Justification:**

The paper presents an interesting and well-supported approach to time-aware video representation learning, and I recommend acceptance.

**Paper Formatting Concerns:**

--

**Quality:**

3

**Strengths And Weaknesses:**

Strengths:

-- Novel and elegant architecture: The idea of linearizing feature trajectories in a latent space is both conceptually appealing and practically effective. The model is simple, efficient (8.7M trainable parameters), and avoids the need for large-scale video model training.

-- Strong empirical results: LiFT outperforms much larger and more complex video models (e.g., VideoMAE, InternVideo 2.5) on the chiral action recognition task across all three datasets. It also shows promise as a plug-in descriptor for general video understanding.

-- Valuable dataset contribution: The CiA dataset is a well-constructed benchmark for evaluating temporal sensitivity in video models. It is diverse, includes both egocentric and exocentric views, and is built from widely used datasets. If released, it would be a valuable resource for the community.

Weaknesses

-- Dataset availability: The CiA dataset is a strong contribution, but its impact depends on public release. The authors should be encouraged to publish the dataset and code for reproducibility.

-- Linearity assumption: While the latent straightening idea is compelling, it would be helpful to see more discussion or visualization of when this assumption holds or breaks down.

---

> ### Author Rebuttal · Authors · 2025-07-29
>
> Thank you for the positive and encouraging feedback! We are encouraged that you found the idea/architecture elegant and the proposed dataset valuable. We address your concerns below.
>
> ---
>
> ### Dataset availability
>
> > Dataset availability: The CiA dataset is a strong contribution, but its impact depends on public release. The authors should be encouraged to publish the dataset and code for reproducibility.
>
> Yes, we completely agree. We will publicly release the *Chirality in Action* dataset along with the code. A preliminary version of the dataset is already attached with the supplementary material.
>
> ---
>
> ### Linearity assumption
>
> > Linearity assumption: While the latent straightening idea is compelling, it would be helpful to see more discussion or visualization of when this assumption holds or breaks down.
>
> We provide qualitative visualization of LiFT’s reconstructed feature trajectories in **Fig 3(a)** in the main paper and **Fig C.2**. in the supplementary material. In general, **LiFT learns a smooth approximation of the feature trajectory corresponding to the visual change** observed in the video. Thus, it works well where there is a clear, consistent visual change (e.g., folding a paper decreases its size, or opening a door gradually increases the angle between the door and the wall, etc.). As shown in Fig C.2. (f), if the visual change is very subtle (e.g., occupies a small part of the image), then it may not be captured by the image features and consequently by LiFT.
>
> On the linearity assumption, the linearity is imposed only in the latent space of the auto-encoder that takes in the sequence of DINOv2 features. Thus, the trajectory of change in the pixel space or in the DINOv2 space can be non-linear, but is mapped to a linearised latent space. This linear direction implicitly encodes the direction of change which is crucial for encoding temporal actions. We will include this discussion in the paper.

---

### Decision · Program_Chairs · 2025-09-17

**Decision:**

Accept (poster)

**Comment:**

The paper received mostly positive reviews with the one borderline reject reviewer unresponsive and failed to put in a final justification. Given this the AC agrees with the other 3 reviewers and recommends acceptance to NeurIPs.